# The cascade of global trade to large climate forcing over the Tibetan Plateau glaciers

Kan Yi[1,9], Jing Meng [2,9], Haozhe Yang[1], Cenlin He[3], Daven K. Henze [4], Junfeng Liu[1], Dabo Guan [5,6], Zhu Liu [6], Lin Zhang [7], Xi Zhu[1], Yanli Cheng[8] & Shu Tao [1]

Black carbon (BC) aerosols constitute unique and important anthropogenic climate forcers that potentially accelerate the retreat of glaciers over the Himalayas and Tibetan Plateau (HTP). Here we show that a large amount of BC emissions produced in India and China—a region of BC emissions to which the HTP is more vulnerable compared with other regions— are related to the consumption of goods and services in the USA and Europe through international trade. These processes lead to a virtual transport pathway of BC from distant regions to the HTP glaciers. From a consumption perspective, the contribution from India to the HTP glaciers shows a rapid increasing trend while the contributions from the USA, Europe, and China decreased over the last decade. International trade aggravates the BC pollution over the HTP glaciers and may cause significant climate change there. Global efforts toward reducing the cascading of BC emissions to Asia, especially the Indian subcontinent, are urgently needed.

[1] Laboratory for Earth Surface Processes, College of Urban and Environmental Sciences, Peking University, 100871 Beijing, China. [2] The Bartlett School of Construction and Project Management, University of College London, London WC1E 7HB, UK. [3] National Center for Atmospheric Research, Boulder, CO 80301, USA. [4] Department of Mechanical Engineering, University of Colorado at Boulder, Boulder, CO, USA. [5] Water Security Research Centre, School of International Development, University of East Anglia, Norwich NR4 7TJ, UK. [6] Department of Earth System Science, Ministry of Education Key Laboratory for Earth System Modeling, Tsinghua University, 100084 Beijing, China. [7] Laboratory for Climate and Ocean-Atmosphere Sciences, Department of Atmospheric and Oceanic Sciences, School of Physics, Peking University, 100871 Beijing, China. [8] Chinese Academy of Meteorological Sciences, 100081 Beijing, China. [9] These authors contributed equally: Kan Yi, Jing Meng. Correspondence and requests for materials should be addressed to J.L. (email: jfliu@pku.edu.cn)

The Himalayas and the Tibetan Plateau (HTP) contain the largest volume of snow and ice outside the polar regions. Snow and glacial melt in the HTP constitute the primary freshwater supply for over 20% of the global population[1]. The rate of warming of the HTP in response to climate change has been reported to be twice the global average[2]. Observations have shown a trend of rapid glacial retreat over the HTP over the past several decades[2–4]. Kraaijenbrink, et al.[5] showed that a global temperature rises of 1.5 °C would lead to a loss of approximately one-third of the present-day glacial ice mass in the HTP by the end of this century. The warming of the HTP and the glacial retreat therein further affect the weather, hydrological cycles, and ecosystems at the regional and global scales, thereby threatening the water availability and food security for hundreds of millions of people[1,6,7].

Black carbon (BC), which is mainly emitted from incomplete combustion, is the second-most important anthropogenic climate forcer in the present-day atmosphere after carbon dioxide[8]. Previous studies have shown that BC aerosols can exert significant impacts on glaciers because of both the heating effects of airborne BC and the albedo effects of deposited BC[2,9–11]. The HTP, which is surrounded by the two most densely industrialized major source regions of BC, namely, South Asia and East Asia, is considered to be more vulnerable to the effects of BC than other regions. Ample evidence shows that the increasing amount of BC being transported into the HTP plays an important role on the observed rapid glacial retreat therein comparable to the effects of greenhouse gases (GHGs)[2,6,12]. The response of surface temperature to per unit radiative forcing from BC on snow and ice was assessed to be two to four times larger than that from $CO_2$[13]. Unlike well-distributed and long-lived GHGs, BC has a heterogeneous spatiotemporal distribution that is extremely sensitive to the emission sites because of a much shorter residence time in the atmosphere. Therefore, tracing the sources of BC being transported into the HTP is essential for providing valuable guidance for an effective mitigation strategy.

Previous studies have made efforts to identify the origins of BC over the HTP during the last decade[14–17]. However, those studies attributed the BC over the HTP to only regions where BC emissions occur during the production of goods and services (i.e., production-based emissions), following atmospheric transport pathways. Meanwhile, BC emissions produced in one region can be further assigned to regions where the related goods and services are ultimately consumed (i.e., consumption-based emissions)[18–20]. The geographical separation of production and consumption following global trade leads to a shift in air pollutant emissions and their associated environmental pressures across regions[21–25]. Therefore, understanding the role of global trade in the distribution of BC emissions and the sharing of responsibility among countries/regions on the BC pollution over the HTP could provide valuable information for the international climate policies making.

To address this issue, we combine four state-of-the-art models to link the BC-related climate forcing over the HTP to different countries/regions as producers and consumers in global trade, respectively. We use the adjoint of the Goddard Earth Observing System (GEOS)-Chem model to identify the locations from which the BC currently situated over the glaciers of the HTP originate and to quantify their relevant emission sensitivities. We further assign the BC emissions in the production process to the final consumers along the supply chain using a multi-region input–output (MRIO) model. The production- and consumption-based emissions are then combined with the emission sensitivities derived from adjoint simulations to estimate the relative contributions of different countries/regions to BC pollutions over the HTP from production and consumption perspectives,

respectively. The corresponding direct radiative forcing (DRF) and snow albedo forcing (SAF) of BC over the HTP glaciers are calculated using a radiative transfer model and a stochastic snow albedo model. In this study, we provide a combined assessment of the sources and radiative forcing of BC over the HTP glaciers from multiple perspectives and reveal the role of global trade in aggravating the BC-related climate forcing over the HTP glaciers. The findings of this study provide valuable implications on the mitigation of the HTP glacial retreat.

## Results

**The atmospheric transport of BC to the HTP glaciers**. The annual-mean mass-burden of BC in the air over the selected HTP glacier regions (defined in Fig. 1) was above $3 \times 10^5$ kg in 2011. An adjoint sensitivity analysis estimates that anthropogenic emissions accounted for 95.4% of the BC arriving in the HTP glacier regions, among which only 4.7% was emitted locally over the HTP. BC emissions produced in India and the rest of China (i.e., outside the HTP) are two major anthropogenic sources that contribute ~30.1% and 16.3% of the annual BC over the HTP glaciers, respectively. Central Asia, Middle Asia, and Southeast Asia, as producers, collectively contribute 9.8% of BC to the atmospheric column over the HTP glacier regions throughout a year.

Figure 2 shows detailed maps of the global anthropogenic BC emission directly contributing to the BC concentrations over the HTP glaciers during different seasons (see Supplementary Fig. 1 for maps of the biomass burning sources). The emission hot spots are located over regions adjacent to the Himalayas, including Nepal and Pakistan. The atmospheric transport pathways of BC to the HTP are characterized by the mid-latitude westerlies and Asian monsoon[16]. Therefore, the sources of BC over the HTP exhibit substantial seasonal and geographical variabilities. A seasonal-mean mass-burden of $4.2 \times 10^5$ kg of BC was estimated over the HTP glaciers during December-January–February (DJF) of 2011, which were transported from world-wide regions following the mid-latitude westerlies due to dry weather conditions and high wind speeds. This amount was nearly halved during the JJA (i.e., June–July–August) because of abundant precipitation and a high BC scavenging efficiency. This seasonal variability in the BC transport to the HTP is consistent with the observations of previous studies[2,26,27]. The contribution of BC emissions produced in China (including the HTP) to the BC pollution over the HTP increased from 17.4% in DJF to 33.4% in JJA. Accordingly, the mass of BC emitted from Southeast China and transported to the HTP in JJA was more than twice that during the other seasons owing to the East Asia summer monsoon.

**Aggravation of BC pollution due to international trade**. Our estimates show that nearly 13% of the ~6.9 Tg of global anthropogenic BC emissions in 2011 were related to traded products through global trade. The interregional flow refers to the transfer of BC emissions between regions where the emissions occur in the production activities and regions where the related goods and services are ultimately consumed. Given the large volumes of interregional flows of air pollutants embodied in global trade, these virtual transport pathways of air pollutants can be orders of magnitude more significant than those of traditional atmospheric transport pathways[23,28].

Figure 3a shows the virtual interregional BC flows embodied in the import/export of goods and services among various countries/regions (see Supplementary Fig. 2 for a map of the defined regions). China, which is a primary BC source region, accounts for 30% of the global anthropogenic emissions. More than 10% of

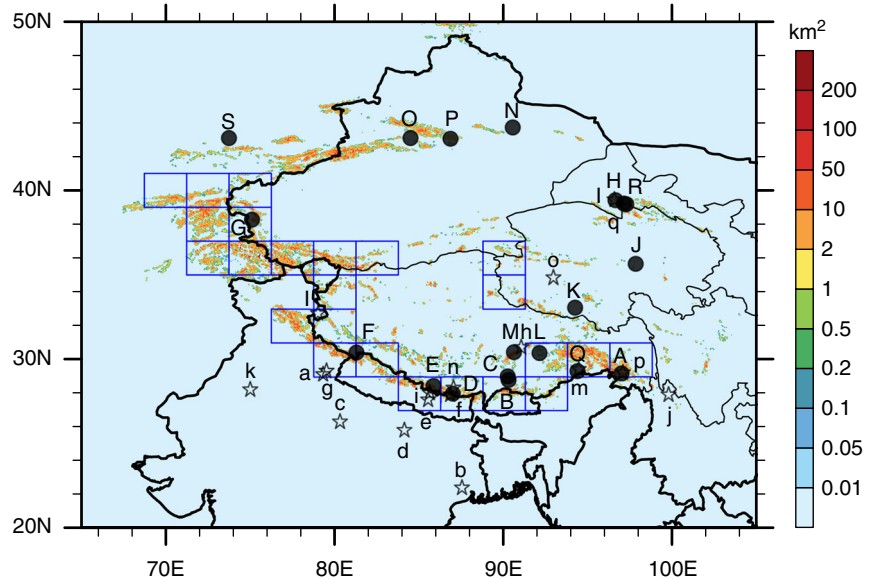

**Fig. 1** The spatial distribution of glacier areas over the HTP and adjacent regions. The rectangles denote the selected HTP glacier regions defined as receptors in the adjoint simulation, where the glacier area in each grid exceeds 2% based on the World Glacier Inventory (WGI)[53]. Black pentagrams are 17 measurement sites of the surface BC concentrations: Nainital (b), Kharagpur (b), Kanpur (c), Gandhi College (d), Nagarkot (e), Langtang (f), Nepal Climate Observatory at Pyramid (NCOP) (g), Manora Peak (h), Nam Co Observational Station (NCOS) (i), Zhuzhang (j), Muztagh Ata (k), Hanle (l), Lutang (m), QOMS (n), Beiluhe (o), Ranwu (p), and QSSGEE (q). Black circles are 20 measurement sites of BC concentrations in the snow: Zuoqiupu (A), Qiangyong (B), Noijin Kangsang (C), East Rongbuk (D), Kangwure (E), Namunani (F), Mt. Muztagh (G), Laohugou #12 (H), Qiyi (I), 1 July glacier(J), Meikuang (K), Dongkemadi (L), La'nong (M), Zhadang (N), Haxilegen River (O), Urumqi Riverhead  and Tianshan Urumpi glacier #1 (P), Miao'ergou #3 (Q), Demula glacier (R), and Muji glacier (S). Please see the Supplementary Tables 2 and 3 for additional details

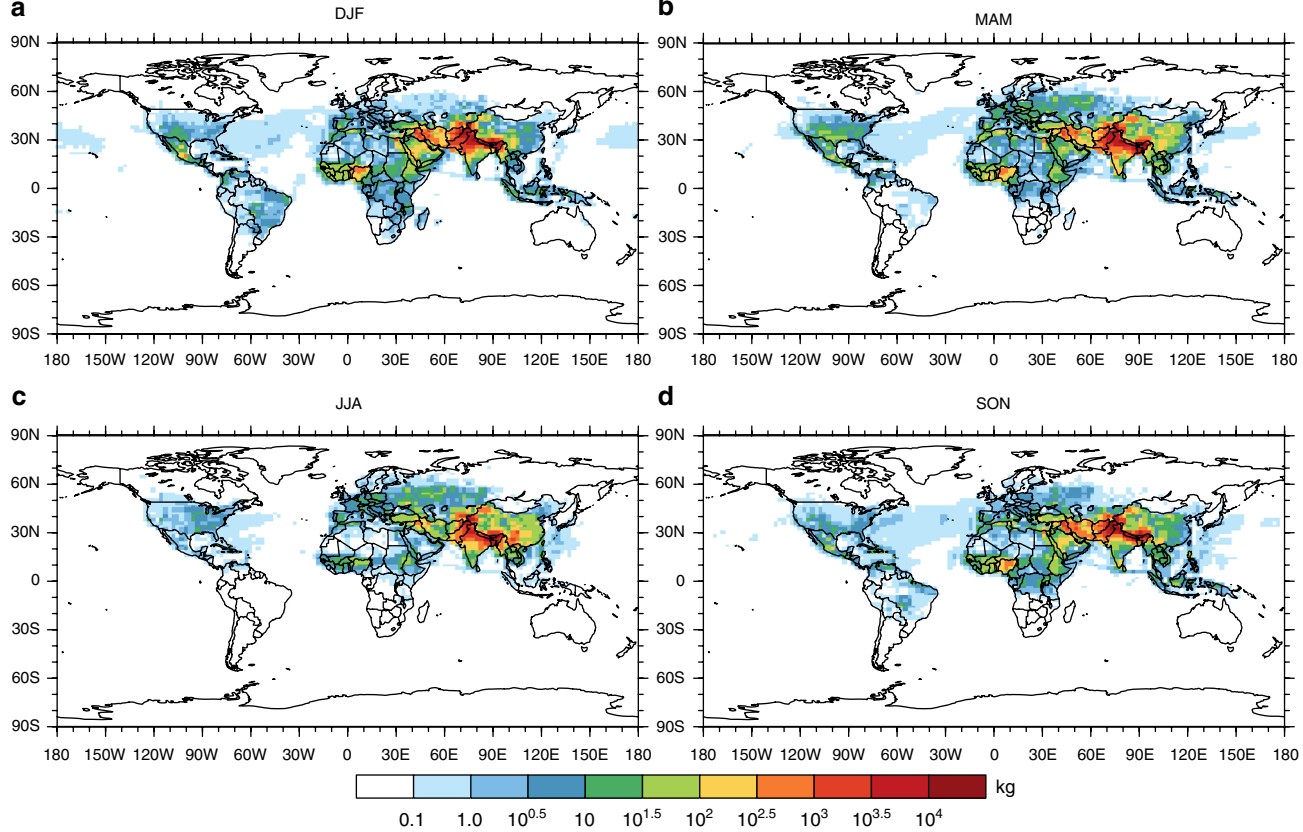

**Fig. 2** Global anthropogenic contributions to BC over the HTP glaciers. The spatial distributions of global anthropogenic BC sources contributed to the BC mass-burden over the selected HTP glacier regions (defined in Fig. 1) during DJF (December–January–February, **a**) MAM (March-April-May, **b**) JJA (June-July-August, **c**), and SON (September-October-November, **d**) in 2011 from the adjoint simulations

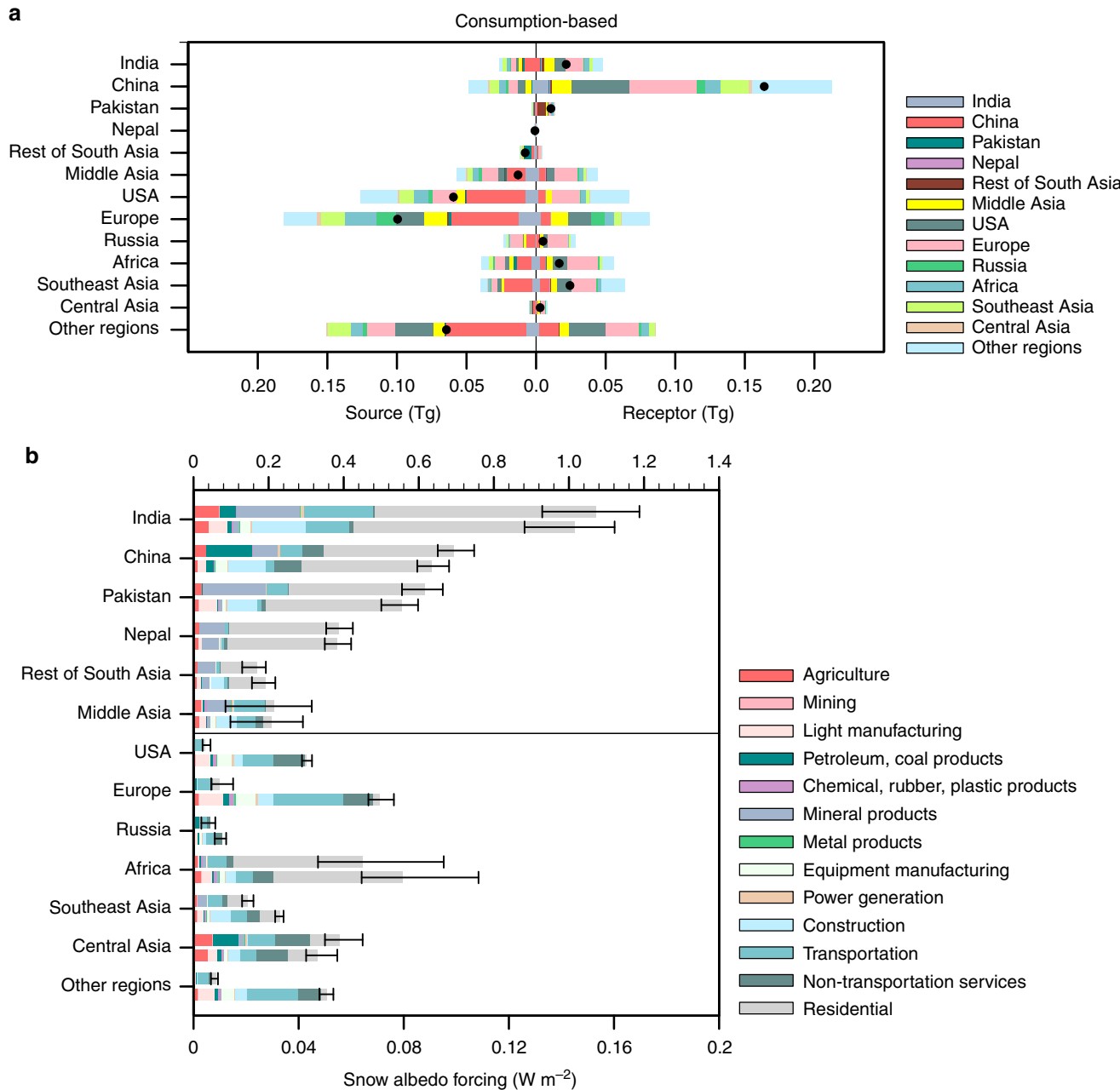

**Fig. 3** BC emissions embodied in global trade and BC radiative forcing over the HTP. **a** Consumption-based interregional flows of BC emissions. Source on the left side refers to BC emissions related to goods and services consumed in the given region that are produced in other regions. Receptor on the right side refers to BC emissions produced in the given region that are related to the consumptions in other regions. Black dots indicate the net interregional flow of BC embodied in global trade as the Receptor minus the Source. **b** Production- (top bar) and consumption-based (bottom bar) snow albedo forcing (W m$^{-2}$) of BC averaged over the HTP glaciers for different regions and sectors. The results for India, China, Pakistan, Nepal, Middle Asia, and the rest of South Asia are displayed according to the x-axis in the top half. The results for the USA, Europe, Russia, Africa, Southeast Asia, Central Asia, and all other regions are displayed according to the x-axis on the bottom half. Error bars denote uncertainty ranges related to inter-annual climate variabilities. Please refer to Supplementary Fig. 4 for the estimation of BC direct radiative forcing at the top-of-atmosphere

its direct emissions (above 0.2 Tg) can be attributed to goods and services consumed by other countries/regions, mainly the USA and Europe (Fig. 3a). Considerable BC emissions in India and Southeast Asia are also related to final consumption in other regions. The consumption of Middle Asia enables the outsource of BC emissions (0.06 Tg) to primarily China and India through the import of goods and services. Africa and Southeast Asia export their goods and services to distant regions like Europe while consume goods and services mainly from China. Even

though the effects of BC emissions produced in the USA and Europe on the HTP glaciers are negligible, the international trade enables shifts of large amount of BC emissions from the USA and Europe to China, India and other Asian regions, and those emissions may be subsequently transported to the HTP (Fig. 3a).

The relative contributions of different regions to the BC over the HTP glaciers from production and consumption perspectives are shown in Supplementary Fig. 3. The USA and Europe play significantly more important roles in the BC pollution over the

HTP glaciers as final consumers than as direct emitters. Our estimation shows that nearly 10% of the BC over the HTP glaciers are related to the emissions embodied in the traded products through global trade. Generally, the global trade leads to a virtual transport pathway of BC from distant regions to the HTP glaciers. This process aggravates the BC pollution towards the HTP glaciers.

**Responsibility for BC climate forcing over the HTP glaciers.** The annual-mean BC DRF at the top-of-atmosphere and the average SAF over the HTP glaciers are estimated to be 1.38 and 3.33 W m$^{-2}$, respectively. Figure 3c shows the BC-SAF over the HTP glaciers contributed by different countries/regions and sectors from production and consumption perspectives. Similar results for the BC DRF are provided in Supplementary Fig. 4. Territorial emissions in India and China are two major contributors to the HTP glaciers, causing BC-SAF of 1.00 and 0.70 W m$^{-2}$ over the HTP glaciers, respectively (Fig. 3c). More than half of the radiative forcing can be attributed to direct residential emissions (e.g., cooking and heating). Mineral products and transportation constitute two major economic sectors that release substantial BC in India. Meanwhile, BC emissions from the production of petroleum, coal and mineral products in China contribute to 0.19 W m$^{-2}$ SAF and 0.08 W m$^{-2}$ DRF over the HTP glaciers (Fig. 3c and Supplementary Fig. 4). Given that substantial territorial emissions are related to the consumption in other regions, the radiative forcing contributed by India and China as final consumers are ~9% lower than as direct emitters (Fig. 3b and Supplementary Fig. 4).

Pakistan and Nepal, which collectively account for only 2.5% of global BC emissions, exhibit large effects on the BC pollution over the HTP glaciers due to their geographical locations. The collective BC-SAF contributed by Pakistan and Nepal, mainly related to cooking and heating and mineral products, is estimated to be 1.00 W m$^{-2}$ and 0.94 W m$^{-2}$ from a production and consumption perspective, respectively (Fig. 3c). Middle Asia, which is also an important contributor to the BC over the HTP glaciers, contributes 0.21 (0.09) W m$^{-2}$ SAF (DRF) as a direct emitter. Furthermore, the USA and Europe overall contribute BC-SAF of 0.11 W m$^{-2}$ to the HTP glaciers as final consumers (Fig. 3b). These effects are nearly 10 times larger than their direct contributions though atmospheric transport pathways. Large components of their consumption-based contributions are derived from transportation and non-transportation services. Moreover, Africa and Southeast Asia contribute 0.11 W m$^{-2}$ BC-SAF to the HTP glaciers as final consumers, which is one and a half times higher than as direct emitters.

**Trends of anthropogenic contributions.** The historical trends of anthropogenic contributions to BC over the HTP glaciers from 2000 to 2014 are estimated from a consumption perspective (Fig. 4). It indicates that the contribution from India to BC over the HTP glaciers has substantially increased by ~40% over the last decade while the USA, Europe, and China all demonstrate a decreasing contribution trend. Specifically, the increasing contribution from India can be mainly attributed to the increasing demand for goods and services in construction and transportation sectors. The significant decrease over China was primarily induced by the mitigation of residential emissions over the last decade. BC emissions related to industrial goods and services consumed in China have increasingly threatened the HTP glaciers from 2000 to 2014 (Fig. 4).

Recently, China has implemented a series of clean air actions to reduce emissions of air pollutants and hence the industrial emission of BC in China has begun to decrease since 2014[29]. Meng et al.[30] showed that some labor-intensive and emission-intensive production activities are moving from China to other developing countries including India because of rising labor costs and industry structure change. BC emissions produced in India that are related to goods and services consumed in other regions have been doubled over the last decade. Evidence has shown that the shifting of polluting industries to regions with more permissive environmental regulations has become a substantial and growing problem[31]. Considering that the HTP glaciers are more vulnerable to BC emitted from India than that from China, we infer that the rise of trade among developing countries and the increasing emissions from India may enhance BC pollutions over the HTP glaciers.

## Discussion

The HTP glaciers, which constitute a crucial water resource, are suffering from rapid retreat due to global warming[4,5]. Both BC and $CO_2$ represent major climate factors; however, while BC persists in the atmosphere for only a few weeks, $CO_2$ can remain in the air for centuries. After depositing on the snow and ice, BC directly warms the cryosphere and accelerates the snow-melting more efficiently than $CO_2$ owing to a positive albedo feedback[13,32]. The mitigation of BC could lead to significant climatic benefits and is therefore urgently needed to preserve the HTP glaciers[33]. Although emissions produced in Asian countries/regions, especially India and China, play crucial roles on the BC pollution over the HTP glaciers due to their high source-receptor sensitivities to the HTP glaciers, our study reveals global trade can further aggravate this problem through a virtual transport pathway of BC from distant regions like US and Europe to the HTP glaciers. Nearly 10% of BC over the HTP glaciers can be attributed to global trade through the relocation of production activities and exchange of commodities. The rapid rise of trade among developing countries contributes to the increase of BC emitted in India and transported to the HTP, which increasingly threatening the HTP glaciers.

Given the significant effects of international trade on BC pollutions over the HTP glaciers, collaborative efforts are needed for an effective mitigation of the BC pollution over the HTP glaciers. Asian countries, especially India and China, should expend substantial efforts, such as improving their energy efficiency, developing clean coal technologies, and promoting clean energy sources, to minimize their domestic BC emissions. Countries/regions that primarily import commodities should care more about the selection of their trading partners or provide assistance in the regulations of trade-related emissions to mitigate corresponding climate effects because the sensitivities of BC over the HTP glaciers to various source regions show large differences.

Present international policies to regulate the transboundary air pollutions, such as the Long-Range Transboundary Air Pollution (LRTAP) Convention[34] and the Malé Declaration on Control and Prevention of Air Pollution[35], consider only the responsibilities of direct emitters and physical transboundary air pollutants. The potential effect of international trade on air pollutions and the shared responsibilities of consumers in this issue are generally ignored by policy-makers. The virtual transport of air pollutants related to the trade of goods and services among countries/regions can be orders of magnitude larger than the typical atmospheric transport[25]. Therefore, future policies should consider the effects of international trade to preserve vulnerable regions like the HTP glaciers.

## Methods
**The origin of BC over the HTP.** The GEOS-Chem model and its adjoint (version 34 with updates to v8-02-01) are used in this study to analyse the origin of BC transported to the HTP. The GEOS-Chem model is a global 3-D chemical

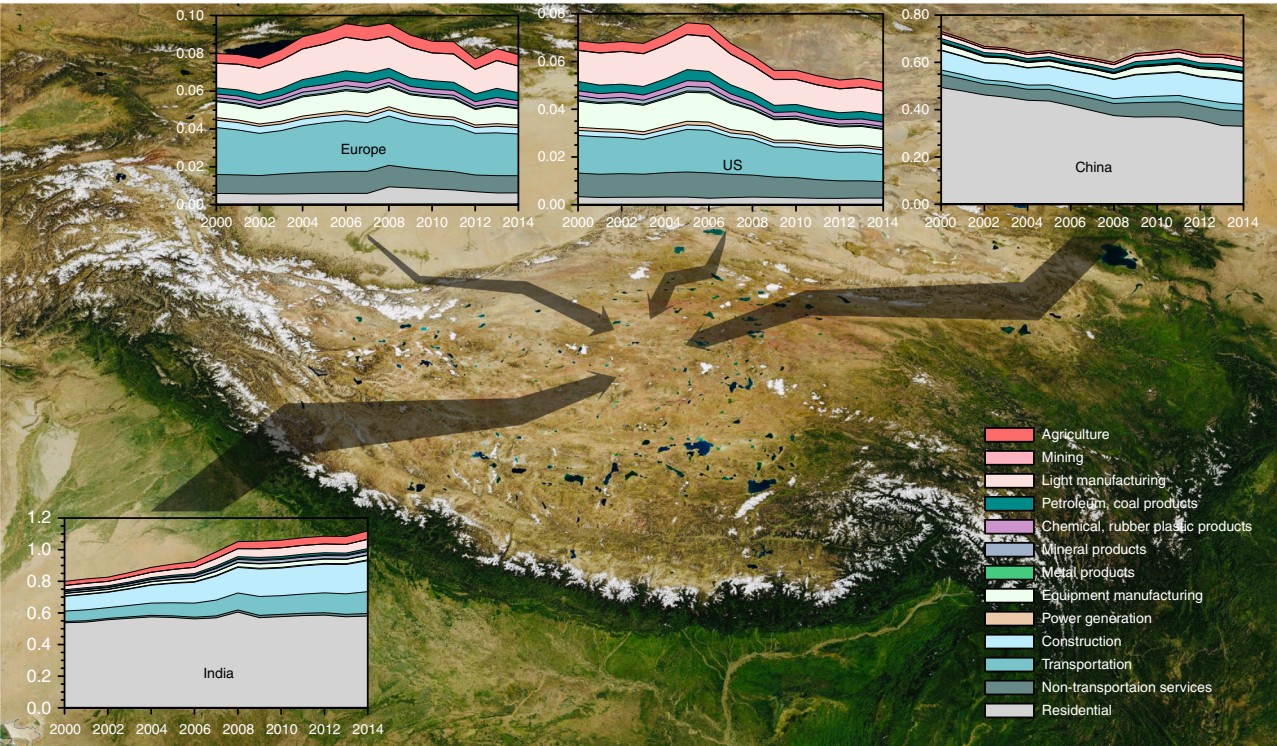

**Fig. 4** Historical trends of consumption-based anthropogenic contributions to BC over the HTP glaciers. Annual-mean contributions ($10^5$ kg) from four different regions (i.e., the USA, Europe, China, and India) and 13 sectors to the BC mass-burden over the HTP glacier regions from a consumption perspective during the period from 2000 to 2014. The thickness of an arrow represents the magnitude of the contribution. The background image is a true-color satellite image from NASA Worldview Snapshots (https://worldview.earthdata.nasa.gov/)

transport model (CTM) driven by assimilated meteorological observations from the Goddard Earth Observing System (GEOS) of the NASA Global Modeling Assimilation Office (GMAO)[36]. The simulation of BC in GEOS-Chem was described in detail by Park et al.[37,38], and its performance has been evaluated extensively in subsequent studies[14,39]. Here, we use the GEOS-5 meteorological data set to drive the model simulation at a horizontal resolution of 2° latitude × 2.5° longitude and a vertical resolution of 47 layers from the surface to 0.01 hPa. In this study, we first conduct five experiments using different emission inventories[40–44] and BC ageing schemes[38,45] to obtain the optical configuration for the adjoint simulation (see Supplementary Table 1). The experiments are conducted for 2006, during which substantial observations over the HTP are available. These experiments are evaluated with in situ observations of BC in the surface air and snow over the HTP and adjacent regions (Supplementary Tables 2 and 3). The GEOS-Chem model exhibits a good performance in simulating the atmospheric transport and deposition of BC over the HTP (see Supplementary Fig. 5 and Supplementary Tables 4–6). The experiment configured with BC ageing scheme from He et al.[45] and anthropogenic BC emission inventory from PKU-BC-Inventory[44] generally exhibits a better performance comparing to other model configurations and hence is further implemented in our adjoint simulations. Supplementary Fig. 6 shows the corresponding spatial distributions of the simulated BC surface concentrations during the different seasons of 2011 in our adjoint simulations. The detailed discussion of model evaluation is provided in the Supplementary Information.

The adjoint model derived from the GEOS-Chem model is an efficient tool for the computation of model sensitivities and inverse problems[46]. This model has been applied to identify the sources[14,47,48] and to constrain the emissions[49–51] of various chemical species (e.g., CO, BC, and $O_3$). Compared with an analysis of the forward model sensitivity, the adjoint approach is far more computationally efficient for source-receptor attribution when the number of sources exceeds the number of receptors[46], which is the case here. The adjoint model calculates the sensitivity (**k**) of the BC mass-burden (**Y**) throughout the whole atmospheric column in the receptor box to the global emissions (**e**). The equation is expressed as follows, where k and e can be resolved at a 2° latitude × 2.5° longitude horizontal resolution:

$$\mathbf{Y} = \mathbf{ke} \tag{1}$$

By multiplying the sensitivity (**k**) with the global emissions (**e**), we can estimate the extent to which the BC emissions from each grid cell contribute to the BC mass-burden in a receptor box (**Y**) following a linear relationship in the BC model

simulation. This approach was previously validated by comparing the adjoint gradients with the forward model sensitivities, which showed good agreement[49,52].

As shown in Fig. 1, we define a large receptor box including 24 model grids, and the glacier area in each grid exceeds 2% based on the World Glacier Inventory (WGI)[53]. The 2° × 2.5° source-receptor sensitivity map of BC in the HTP derived from the adjoint analysis is disaggregated to 0.1° × 0.1° using high-resolution emission inventory as a proxy. The emission intensity at a 0.1° × 0.1° resolution is obtained from the PKU-BC-Inventory[44] (Supplementary Fig. 7). This approach is based on a basic principle that regions emitted more BC may have more contributions to the HTP and should share more responsibilities. It considers the sub-grid distribution of BC emissions and hence covers the shortage of the model resolution more or less. The adjoint simulations are conducted for all months from 2007 to 2011 with BC emissions fixed in 2011. Each case spans three weeks of the corresponding month and the model response (**Y**) is defined for the last week. Our analysis focuses on the results of 2011 to ensure consistency with the multi-region input–output (MRIO) analysis. Meanwhile, we use the multiyear result to estimate the uncertainty range associated with the inter-annual variabilities in the climate and the corresponding changes in atmospheric transport pathways. To evaluate the performance of adjoint simulations, we further conduct two sensitivity simulations using the GEOS-Chem forward model in which the consumption-based BC emissions from China and India are removed, respectively (Supplementary Fig. 8).

**Emissions related to the final consumers**. Consumption-based emissions reallocate the pollutants released throughout the supply chain to the final consumers of goods and services. Based on detailed data of the economic structure describing the product exchanges within and among regions, the MRIO quantitatively assigns emissions produced by different sectors in one region according to the final demand for goods and services.

For an MRIO analysis with $M$ regions and $N$ sectors in each region, the total output of industry $i$ in region $r$ ($x_i^r$) can be represented as follows:

$$x_i^r = \sum_{s=1}^{M} \sum_{j=1}^{N} a_{ij}^{rs} x_i^r + \sum_{s=1}^{M} y_i^{rs} \tag{2}$$

where $a_{ij}^{rs}$ ($r, s = 1, 2, …, M; i = 1, 2, …, N$) is defined as the proportion of the input from sector $i$ in region $r$ to the production of one unit of output from sector $j$ in region $s$, $y_{ij}^{rs}$($r, s = 1, 2, …, M; i = 1, 2, …, N$) represents the finished goods by sector $i$ produced in region $r$ and consumed in region $s$. Equation (2) can also be

expressed in matrix form as

$$\mathbf{X} = \mathbf{A}\mathbf{X} + \mathbf{Y} \qquad (3)$$

and further transformed as follows:

$$\mathbf{X} = (\mathbf{I} - \mathbf{A})^{-1}\mathbf{Y} \qquad (4)$$

where $(\mathbf{I} - \mathbf{A})^{-1}$ is the Leontief inverse matrix. Given the output in each region, the region- and sector-specific production related to the consumption activity of a given region $r$ can be calculated as follows:

$$\mathbf{X}_{com}^{\prime r} = (\mathbf{I} - \mathbf{A})^{-1}\mathbf{Y}^{\prime r} \qquad (5)$$

where $\mathbf{Y}^{\prime r}$ is a vector for the region $r$ that includes the final consumption produced domestically ($\mathbf{Y}^{rr}$) and the final consumption imported from other regions ($\mathbf{Y}^{s,r}$, $s \neq r$), and $\mathbf{X}_{com}^{\prime r}$ is a vector of the total production over multiple regions and multiple sectors related to the consumption in the region $r$. By incorporating a vector of the emission intensity ($\mathbf{F}$) representing the region- and sector-explicit emissions embodied by one unit of product derived from a production-based emission inventory, the consumption-based emission ($\mathbf{E}^{\prime r}$) in the region $r$ is constructed as follows:

$$\mathbf{E}^{\prime r} = \mathbf{F}(\mathbf{I} - \mathbf{A})^{-1}\mathbf{Y}^{\prime r} \qquad (6)$$

The primary MRIO analysis is conducted in 2011 using the latest economic data from version 9 of the Global Trade Analysis Project (GTAP)[54]. The GTAP database covers 140 regions (Supplementary Data 1) and 57 industrial sectors (Supplementary Data 2) in addition to monetary flows between industrial sectors and regions. To be consistent with the adjoint analysis, the global production-based BC emission inventory is derived from the PKU-BC-Inventory[44]. This inventory, which covers 64 fuel combustion processes and 14 industrial processes throughout 233 countries/territories, is converted into 140 regions and 57 sectors to be consistent with the GTAP database. The details regarding a reconciliation of the data between the emission inventories of sectors used for input–output analysis and gridded emission inventories used for atmospheric CTMs can be found in Meng et al.[21]. Supplementary Figure 9 shows the result of the production- and consumption-based annual BC emissions of different regions in 2011.

In addition, we conduct a time-series MRIO analysis to examine the historical trend of BC emissions embodied in global trade from 2000 to 2014 using the World Input-Output Database (WIOD)[55]. The WIOD data set covers 28 European countries and 15 other major countries throughout the world[55]. The GTAP data set has a more detailed classification for countries/regions that could help us distinguish the contributions from Pakistan, Nepal, and other Asian regions, but it does not provide continuous economic data. Since we only use the WIOD database to analyse the trends in the consumption-based emissions of several major regions (i.e., the USA, Europe, China, and India), the differences between the two databases will not cause any inconsistence in our results. The production- and consumption-based emissions in different countries/regions are further combined with the source-receptor emission sensitivity derived from adjoint simulations to estimate the relative contributions of different regions to the BC in the HTP from production and consumption perspectives. To distinguish the effects of consumption-based emissions from that of climate variabilities on the historical trends, the source-receptor emission sensitivity is fixed in the year 2011 for both the primary and time-series analysis.

**BC radiative forcing.** The Fu-Liou-Gu (FLG) radiative transfer model (RTM)[56,57] is used to calculate the instantaneous clear-sky BC direct radiative forcing (DRF) in the atmosphere. In the FLG RTM, the delta-four-stream approximation for solar flux calculations is combined with the delta-two/four-stream approximation for infrared flux calculations to assure both the accuracy and the efficiency. The solar (0–5 μm) and infrared (5–50 μm) spectra are divided into 6 and 12 bands, respectively, based on the locations of absorption bands. The correlated $k$-distribution method is used to differentiate the gaseous absorption lines. The spatial and temporal distributions of the BC concentrations and the optical depth used within the RTM are obtained from the GEOS-Chem outputs. The meteorological inputs are taken from the GEOS-5 assimilated fields.

A stochastic snow albedo model (SSAM)[58–60] is used to calculate the BC-snow albedo forcing (SAF). This model explicitly simulates multiple BC particles both stochastically and internally/externally mixed with various shapes of snow grains. Specifically, the light absorption and scattering (i.e., single scattering properties) of the BC-snow mixtures are calculated using an improved geometric-optics surface-wave (GOS) approach[61–63]. The single scattering properties of an ensemble of randomly oriented snow grains (mixed with BC) are further averaged over all directions. The underlying ground albedo is obtained from GEOS-5 meteorological fields. The snow albedo is computed via the adding/doubling radiative transfer method using the single scattering properties of the BC-snow mixtures. We fully consider different snow grain shapes (sphere or spheroid), BC-snow mixing states (internal or external), and BC coating states (coated or uncoated) in our estimation, and the mean value is used in the final report. The BC-induced snow albedo reduction is calculated as the difference in the snow albedo both with and without BC. This albedo reduction is further coupled with the GEOS-5 incoming solar radiation field to obtain the surface radiative forcing following Wang et al.[64].

The GEOS-Chem-simulated BC concentrations in the air and snow are input into the FLG RTM and SSAM. The global DRF at the top-of-atmosphere (TOA) and the BC-SAF estimated in this study are 0.54 and 0.07 W m$^{-2}$, respectively, which fall within the ranges published in the Intergovernmental Panel on Climate Change (IPCC) Fifth Assessment Report (0.05–0.8 and 0.02–0.09 W m$^{-2}$, respectively)[13]. The spatial distributions of BC-related DRF and SAF are shown in Supplementary Fig. 10. These radiative forcing (i.e., DRF and SAF) of BC are averaged over the HTP glacier region selected in this study and further assigned to different regions and sectors according to their relative contributions to the BC concentrations from production and consumption perspectives.

Supplementary Fig. 11 illustrates the methodological framework for integrating the multi-models. Related uncertainties and limitations are discussed in the Supplementary Information.

## Data availability

The glacier area dataset is obtained from the World Glacier Inventory (WGI, http://nsidc.org/data/glacier_inventory/). The anthropogenic emissions of BC is available from PKU-BC-Inventory (http://inventory.pku.edu.cn/). The economic input-output data are available from version 9 of the Global Trade Analysis Project (GTAP, https://www.gtap.agecon.purdue.edu/) and the World Input-Output Database (WIOD, http://www.wiod.org/database/wiots16). Maps in Figs. 1, 2 and Supplementary Figs. 1–3 and 6–10 are generated by the NCAR Command Language (Version 6.6.2) (https://doi.org/10.5065/D6WD3XH5). Supplementary Data 1 and 2 are the definition of regions and sectors, respectively. The source data underlying Figs. 3–4 and Supplementary Figs. 3 and 4 are provided as a Source Data file. The datasets generated during this study are available in the figshare repository with the identifier https://doi.org/10.6084/m9.figshare.7806827.

## Code availability

All computer codes generated during this study are available in the figshare repository with the identifier https://doi.org/10.6084/m9.figshare.7806827.

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

## Acknowledgements

This work was supported by funding from the National Natural Science Foundation of China under award nos. 41571130010, 41671491, 41821005, 41629501, 71533005 and 71874097; National Key Research and Development Program of China 2016YFC0206202; Chinese Academy of Engineering (2017-ZD-15-07); NOAA grant NA16OAR4310113, British Academy (NAFR2180104, NAFR2180103), the UK Natural Environment Research Council (NE/N00714X/1 and NE/P019900/1), the Economic and Social Research Council (ES/L016028/1), the Royal Academy of Engineering (UK-CIAPP/425), and the 111 Project (B14001). The National Center for Atmospheric Research is sponsored by the National Science Foundation. We acknowledge the use of imagery from the Worldview Snapshots application (https://wvs.earthdata.nasa.gov/), part of the Earth Observing System Data and Information System (EOSDIS).

## Author contributions

J.L., K.Y., and J.M. designed the study. K.Y. conducted GEOS-Chem and its adjoint simulations. D.K.H. evaluated application of the adjoint model. J.M. and H.Y. conducted the multi-regional input–output analysis. C.H. calculated BC radiative forcing. X.Z., S.T., and C.H. provided emissions data. K.Y., J.L., J.M., and C.H. interpreted data. J.L. coordinated and supervised the project. K.Y., J.M., and J.L. led the analysis and writing. K.Y., J.M., H.Y., C.H., D.K.H., J.L., D.G., Z.L, L.Z., X.Z., Y.C., and S.T. contributed to the writing of manuscript.

## Additional information

**Competing interests:** The authors declare no competing interests.

