## [Peer Review File · Nature Communications]

Reviewers' comments:

Reviewer #2 (Remarks to the Author):

Please refer to the attached review document.

Reviewer #3 (Remarks to the Author):

review of Yi et al.

“How do global trade cascade to huge climate forcing over the third pole glaciers.”

Summary - This paper calculates forcing by black carbon emitted from different kinds of trade and different producers and consumers. To do this, the authors connected an input-output model with an atmospheric model that simulates transport and deposition of black carbon on the glaciers. Then each of the estimated forcings is attributed to its cause, i.e. internal activity, or international trade.

Reaction – The models used for the analysis are well accepted and so the result appears valid. However the presentation doesn't provide new insights and yet it is sensationalized and that leads to a negative impression.

It is known that modeled black carbon forcing is large over Himalayas. It is known that especially China has a big role in trade and some papers on pollutant attributed to trade have been published, including the authors' own work. This paper does add quantitatively to the estimates. But for many countries all the methods (Production, Consumption, Income) of looking at the contribution to BC forcing are similar. This is because many of the sources are internal to countries and not related to trade as the authors point out. They also point out “the percentage of global BC emissions embodied in the global trade is nearly two times smaller than that of CO₂” so CO₂ is embedded in trade more than BC. Why then is the paper title emphasizing trade? Perhaps it should emphasize the internal activities, instead!

Then, the title again states there is a “huge climate forcing.” This forcing isn’t greater than that shown in other papers. And it could not be as the model is not new. What is huge? Is it bigger than other forcings? Again the science appears supportable, but the dramatic statement, and uncritical policy statements like “regions... should be cautious about their direct emissions” suggest a lack of sophistication.

Scientific comments: (Addressing any one of these could have made the analysis more novel.)

1. The transport model used here is coarse resolution and has difficulty representing transport to Himalayas.
2. Black carbon forcing is caused by change in snow albedo, snow albedo depends on dust and even little dust can reduce the forcing of black carbon. The dust treatment isn’t discussed.
3. The net forcing is not caused by only black carbon but also organic carbon, sulfate, ozone in addition to CO₂. Is black carbon the dominating forcing? Is it acceptable to disregard the other pollutants? Any change in activity, or trade etc, would affect all of these.

Reviewer #4 (Remarks to the Author):

The title of this manuscript was interesting that linked the glaciers retreat with global trade. Many recent studies indicated the black carbon (or other pollutions) originated from South Asia and the rest regions outside of the Third Pole (TP) could transport to the plateau and impacted on the local climate, using different approaches including monitoring network, geochemical evidence and combination of modeling technology. More results showed India should take responsibility for glaciers melting in the TP induced by anthropogenic BC, however, ignored the potential transfer BC emission from USA and Europe through global trade. I generally agreed the authors’ viewpoints and thought the results were novel. Furthermore, the manuscript needs substantial modification until reaching to the criteria.

In the title referred to the “The third Pole”, this domain was a great region not only involving the Tibetan Plateau and Himalayas, but also including Tianshan Mt., Qilian Mt., Kunlun Mt. and the others mountain regions over the Xinjiang, Yunnan, and Sichuan in China. Actually, HTP ≠ Third Pole. So the authors should consider more glacier areas as showed in Fig. 1 blue rectangles.

The method was not clearly written. Specifically, how did you obtain the seasonal mean BC concentration contributed by consumption-based from India and China, as showed in Fig. s5. I suggested to provide a framework in SI to help reader understanding the steps.

GEOS-CHEM was used to calculate the contribution from the source region, but its resolution was very coarse with 2*2.5°(may be more). As showed in Fig.2, I saw too much mass burden over the

Southern TP and Himalayas, however, actually there was no or few emission derived from the local activities. I trust such “biases” was induced by the coarse resolution. The authors mentioned a process that transfer the grid of $2^{\circ} \times 2.5^{\circ}$ to $0.1^{\circ} \times 0.1^{\circ}$, please provide the details. Did you consider any other factor such as terrain effect in the transfer process? Otherwise the interpolation of such downscale did not make sense. Another question was the connection between the modeling approach of MRIO and GEOS-CHEM, please clarify.

The authors suggested that the virtual transport pathway was more efficient than the traditional atmospheric pathway. Does that mean the traditional pathway might be underestimated? If it was true, how was the updated contributed rate (the traditional atmospheric pathway were approximately 68.9% and 16.3% from South Asia and China respectively) once the virtual pathway was considered.

The BC inventory was a probably uncertainty factor both in GEOS-CHEM and MRIO. I found only a dataset called “PKU-BC-inventory” was input. Even though the data was well agreement with observation, it was preferable to compare with another dataset of inventory applied in the model. In this way, the results trend to be more solid.

For model evaluation, the current collections from previous studies were not the updated data set (most data were for 2004-2006 in Table S3, S6). In fact, recent years many publications reported the observations in terms of BC concentrations both in the atmosphere and in snow as well as its albedo effect over the Third Pole.

The most updated data and publications must be considered, such as:

Li X, et al. (Light-absorbing impurities accelerate glacier melt in the Central Tibetan Plateau. *Science of the Total Environment*, 2017, 587: 482-490.) reported the data in the Central of Tibetan Plateau.

Niu, H., et al. (In-situ measurements of light-absorbing impurities in snow of glacier on Mt. Yulong and implications for radiative forcing estimates. *Science of The Total Environment*, 2017, 581, 848-856.) reported the data in the southeastern Tibetan Plateau.

Zhang, Y., et al. (Light-absorbing impurities enhance glacier albedo reduction in the southeastern Tibetan plateau. *Journal of Geophysical Research: Atmospheres*, 2017, 122. Doi: 10.1002/2016JD026397.) reported the data in the southern Tibetan Plateau.

Zhang, Y., et al. (Characteristics of black carbon in snow from Laohugou No. 12 glacier on the northern Tibetan Plateau. *Science of the Total Environment*, 607, 1237-1249.) reported the data in the northeastern Tibetan Plateau.

Zhang, Y., et al. (Black carbon and mineral dust in snow cover on the Tibetan Plateau. *Cryosphere*, 2018, 12(2), 413-431.) reported more data of BC-in-snow cover the region of Tibetan Plateau.

Zhang Y. L. et al. (Research progress of light-absorbing impurities in glaciers of the Tibetan Plateau and its surroundings (in Chinese). *Chin. Sci. Bull.*, 2017, 62(35): 4151-4162. Doi: 10.1360/N972017-00505)) provided a review BC and other LAIs in the Third Pole.

Chen X. et al. (Concentration, temporal variation, and sources of black carbon in the Mt. Everest region retrieved by real-time observation and simulation. *Atmos. Chem. Phys.*, 2018, 18, 12859-

12875. Doi: 10.5194/acp-18-12859-2018) reported two year BC in the atmosphere on Everest Station.

Minor suggestions for references. In Line 71, citations 2, 6 indicated the increased BC could induce rapid glacier melting, but did not referred or showed the evidence the BC being transported into the HTP. One study based on the ¹⁴C isotope technology (Li C. et al., Sources of Black Carbon to the Himalayan-Tibetan Plateau glaciers. Nature Communications, 2016, 7: 12574. Doi: 10.1038/ncomms12574.) has provided the geochemical evidence. Line77, "Various approaches..." were not various, actually all were based on modeling technology.

Response to Anonymous Reviewer 2:

(Note: Reviewer comments are listed in blue, and responses to reviewer comments are in black. Pasted text from the new version of the paper is in italics.)

Suggested revision: Major revision and resubmission

General Impression

This paper attempts to assess the responsibility for the loss of glaciers in the Himalayan from black carbon emissions only. The authors combine a large set of concepts and tools to achieve this. Some of these major concepts are

- use of an atmospheric pathway model to assess how black carbon emissions are transported to the Himalayan mountains, and where they originate from
- the separation of the radiative forcing process induced by black carbon emissions only from all other radiative forcing (especially the radiative forcing from CO₂)
- consideration of the temporal perspective of short-lived black carbon emissions and long-lived CO₂ emissions, and the varying effects of these on glacier loss
- the use of two multi-regional input-output models to assess the implications that trade has on the emissions of black carbon in those regions that contribute to the black carbon emissions experienced in the Himalayan
- consideration of a temporal perspective to assess the historical development of black carbon emissions

The authors had to make certain assumptions in order to combine these different aspects. In my view, the way in which these concepts were combined weakens the reliability of the study. In my opinion, the study would greatly benefit from reducing the scope in order to provide more thorough linkages between the above-mentioned concepts.

The paper structure needs improvement. Certain concepts are sometimes not introduced at the time they are used in the discussion, and assumptions about the modelling approach are discussed in later parts of the paper. This weakens the previous discussion in several instances. A solid set-up of the paper with a narrower focus and key take-away messages would improve the work. For example, the time series consideration is a side-note in the paper, but it actually sparks questions regarding the compatibility of the models in different years.

Questions that I would have liked to see addressed by this paper are:

- if the glaciers lose one third of their mass by the end of the century, how much less water will be available where?
- how much of this loss can be attributed to the black carbon emissions?
- if black carbon emissions were reduced by say 50% by 2030 (for example), how much of the glacier mass is saved?

- how much glacier mass would we lose by the end of the century if black carbon emissions could be stopped all completely? How much more water would remain for the people in the area compared to the business as usual case?

We really appreciate the reviewer's thoughtful and valuable comments. In this study, we systemically explored the contributions from different countries/regions to the black carbon (BC) forcing over the Himalayas and the Tibetan Plateau (HTP) using an adjoint inverse model combined with multi-region input-output analysis. The HTP contains the largest volume of snow and ice outside of the polar regions and has been identified to be critical in regulating the Asian hydrological cycle and climate system (Immerzeel et al., 2010; Pritchard, 2017a). The glaciers over the HTP provides fresh water to billions of people as a water storage tower for South Asia, Southeast Asia and East Asia. However, the glaciers over the HTP were observed to be in rapid retreat trend over the past several decades associated with climate change. The warming of the HTP and the glacial retreat therein further affect the weather, hydrological cycles, and ecosystems at the regional and global scales, thereby threatening the agriculture, hydropower and even national security for the developing countries in the region. Source attributions of climate forcing over the HTP is extremely necessary for an efficient mitigation of the rapid glacier retreat therein.

Here we focus on the BC because:

- (1) BC was regarded as the second most important factor causing the fast melting of HTP glaciers (Bond et al., 2013; Qian et al., 2015). The Atmospheric Brown Cloud project showed that the melting of the Himalayan glaciers is related to enhanced heating from BC aerosols and GHGs of 0.25K per decade, from 1950 to present, of which the BC associated heating is 0.12K per decade (Ramanathan et al., 2005; Ramanathan et al., 2007).
- (2) The property of BC is quite different from that of CO₂. BC persists in the atmosphere for only a few weeks while CO₂ can remain in the air for centuries. CO₂ is nearly uniformly distributed in the atmosphere after it was emitted from human activities. The amount of CO₂ over the HTP relies more on the total amount of CO₂ emitted from global-wide than the exact locations of these emissions. On the contrary, BC has a heterogeneous spatiotemporal distribution, which means that the BC over the HTP is extremely sensitive to the emission sources. Therefore, tracing the sources of BC being transported into the HTP is more essential for an effective mitigation strategy over the HTP.
- (3) The mitigation of BC could lead to immediate climatic benefits that is urgently needed to preserve the HTP glaciers. BC directly increases the net solar radiation by reducing surface albedo after deposits on the snow and ice, while CO₂ first warms the air by absorbing more longwave radiation then warms the surface and melts the snow by air-surface heat transfer. Therefore, the response of surface temperature to per unit radiative forcing from BC on snow and ice was assessed

to be two to four times larger than that from CO₂ (Qian et al., 2011; Stocker et al., 2013). More importantly, BC deposited on the snow and ice would mostly accumulate in the snow-pack instead of being washed away during the snow melting, which is regarded as the post-depositional process (Xu et al., 2012b). This process enriches BC concentrations on the snow that may significantly accelerate the melting of HTP glaciers.

Moreover, we agree with the reviewer that the quantification of glacier mass loss associated with BC is quite an interesting topic. Previous studies have taken effort to explore this issue. For example, Qian et al. (2011) designed a series of numerical experiments with a global model to assess the relative impacts of anthropogenic CO₂ and BC on the Asian hydrological cycle. They indicated that BC-in-snow increases the surface air temperature by around 1.0 °C averaged over the TP and reduces spring snowpack over the TP more than pre-industrial to present CO₂ increase and BC in the atmosphere. Yasunari et al. (2010) estimated the impacts of BC in snow on the runoff from Himalayan glaciers by applying simple numerical experiments with a glacier mass balance model. It showed that snow albedo reductions associated with BC would lead to runoff increases of 70-204 mm of water, which is equivalent of 11.6-33.9% of the annual discharge of a typical Tibetan glacier. Menon et al. (2010a) indicated that snow/ice cover decreased by ~0.9% over the Himalayas from 1990-2000 in which the contribution of the enhanced Indian BC to this decline is ~36 %. As for the future projection, Wang et al. (2012) predicted that BC emissions in China in 2050 will be 920–2183 Gg/yr under various scenarios. Following the four CMIP5 RCPs, Bauer et al. (2013) simulated BC deposition fluxes and radiative forcing on the Antarctic, Greenland, and the Himalayas for the 2006-2100 period, which indicated further dramatic advances of pollution to the Tibetan Plateau. Kraaijenbrink et al. (2017) showed that a global temperature rise of 1.5 °C would lead to a loss of approximately one-third of the present-day glacial ice mass in the HTP by the end of this century.

Though the topic is very important, we also agree with the reviewer that this study should reduce the scope. Therefore, this study focuses on providing a combined assessment of tracing the sources and radiative forcing of BC over the HTP glaciers from not only the local direct emissions, but also the global supply chain.

Furthermore, following the reviewer's suggestion, we have improved the **introduction, methods and discussion**. Specifically, we have added more details to illuminate the value of this study in the **introduction**. For the **discussion**, we have removed the income-based part and focused our discussion on the production and consumption perspectives. The time series analysis and its implications have been discussed more thoroughly. The structure of the **discussion** has been significantly improved then. The key take-away message of this study is that our analysis reveals an aggregating effect of global trade on the climate forcing over the HTP, which may

increasingly threaten the glacier retreat therein. In the revised manuscript, we have further clarified the text to facilitate understanding.

All the questions raised by the reviewer have been properly addressed. We believe it substantially helps to improve our manuscript by addressing these issues. Please see our response to each comment below.

Detailed feedback

Abstract

The statement "The USA and Europe transfer BC emission to Asian regions" is a bit unclear to me. The following sentence hints that this statement is related to embodied BC emissions through trade. But I think that distinction should be made clear in the abstract: There are direct BC emissions from India and China, and a number of these are indirectly driven by consumption in the US and China. Having said that, the abstract currently covers direct emissions in its first part, then indirect BC emissions accounted to the USA and Europe, and then direct emissions from India, which are again linked to consumption. I would suggest to stick to a linear narrative as follows: Cover the physical effects (as you did in the beginning), then the direct emitters and they are close by (India and China), and then cover the aspect that from a consumption perspective, and that emissions substantially driven by trade partners.

We thank the reviewer for your constructive comments. We have followed the reviewer's suggestion and reorganized the abstract, see details below:

"...Here we show a large amount of direct BC emissions from India and China, where the HTP is more vulnerable to, are indirectly driven by the USA and Europe through the importation of commodities. These processes lead to a virtual transport pathway of BC from remote regions to the HTP glaciers. From a consumption perspective, the contribution of BC emissions from India to the HTP glaciers shows a rapid increasing trend while the contributions from the USA, Europe, and China decreased over the last decade. Our results reveal that international trade aggravates the BC pollution over the HTP glacier regions and may cause significant climate change there. Global efforts toward reducing the cascading of BC emissions to Asia, especially the Indian subcontinent, are urgently needed."

Line 164

The authors are referring to Figures 3a and b here. These figures show consumption-based and income-based responsibility. At this point in the paper, the concepts of consumption-based, and particularly income-based emissions are assumed knowledge by the reader. I am familiar with MRIO analysis, but I am not aware of a generally agreed concept of income-based emissions. The supplementary information covers the Ghosh-model, which could be used to calculate income-based effects. I am however not sure how income-based emissions would be calculated. Also, the Ghosh-model has received substantial criticism towards its quantitative applicability. Please provide

an extensive discussion of the concept and value of income-based emissions. Further, the authors do not refer to these two different accounting methods (consumption-based and income-based) in the main text. I believe that the main text and Figures 3a and b can be more aligned.

We appreciate the reviewer's comments considering the debate on the income-based effects and Ghosh-model. possible effects from cross-hauling. We do agree with the reviewer that the Ghosh-model used to calculate income-based emissions may receive substantial criticism. We followed the reviewer's suggestion and removed the accounting of income-based part in discussion. In the revised manuscript, we have focused our discussion on the production and consumption perspectives and improved the structure of the manuscript significantly. For the concept and calculation of income-based emissions, we follow the previous studies (Liang et al., 2017; Steining et al., 2016). The income-based accounting method assigns direct geographic emissions (i.e., production-based emissions) to primary suppliers of labour forces and capital. This measure accounts for all the emissions generated downstream in its supply chain for a certain product until delivery to final demand. Different from the production- and consumption-based accounting methods, the income-based metrics measure the emissions enabled by primary suppliers and that are required to generate a country's income through wages, profits and rents. Through an income-based approach, the benefit from the generation of the emissions is passed to suppliers in the form of income, which may also have the responsibility for the emissions generated. The Figure 3 and other relevant results have been revised correspondingly.

Additionally, we have reorganized the **Materials and Methods** section to provide more details about the consumption-based accounting method. Please refers to the revised manuscript for more details.

In main text:

“Aggravation of BC pollution over the HTP glaciers due to international trade

Our estimates show that nearly 13 % of the ~6.9 Tg of global anthropogenic BC emissions in 2011 were embodied by consumption-based accounting methods within global trade. Interregional BC flow refers to BC emissions from one country/region that cannot be assigned to local consumers but are instead embodied by the interregional exchange of commodities with other countries/regions. Given the large volumes of interregional flows of air pollutants embodied in global trade, the efficiencies of these virtual transport pathways of air pollutants are orders of magnitude higher than those of traditional atmospheric transport pathways^{23,28}.

Figure 3a shows the virtual interregional BC flows embodied in the import/export of goods and services among various countries/regions (see Supplementary Figure 9 for a map of the defined regions). China, which is a primary BC source region, accounts

for 30 % of the global anthropogenic emissions. More than 10 % of its direct emissions (above 0.2 Tg) can be attributed to products and services consumed by other countries/regions, mainly the USA and Europe (Figure 3a). Considerable BC emissions in India and Southeast Asia are also related to final consumption in other regions. The consumption of Middle Asia enables the downstream transport of BC emissions (0.06 Tg) to primarily China and Europe. Africa and Southeast Asia export their products and services to remote regions like Europe while consume products and services mainly from China. Even though the effects of direct BC emissions from the USA and Europe on the HTP glaciers are negligible, the international trade transfers large volumes of BC emissions from the USA and Europe to China, India and other Asian regions, and those emissions may be subsequently transported to the HTP (Figure 3a).

The relative contributions of different regions to the BC over the HTP glaciers from production- and consumption-based perspectives are shown in Supplementary Figure 10. The USA and Europe play significantly more important roles in the BC pollution over the HTP glaciers as final consumers than as direct emitters. Our estimation shows that nearly 10 % of the BC over the HTP glaciers are embodied by interregional flow within global trade though the importation of commodities. Generally, the global trade leads to a virtual transport pathway of BC from remote regions to the HTP glaciers. This process aggravates the BC pollution towards the HTP glaciers. ”

In Materials and Methods

“Emissions embodied in global trade

Consumption-based emissions refer to pollutants released throughout global trade as a result of the consumption of goods and services. Here we use an MRIO model to estimate global BC emissions embodied in global trade. Based on detailed data of the economic structure describing the product exchanges within and among regions, the MRIO quantitatively assigns emissions produced in one region to another region throughout global trade and determines where goods and services are ultimately consumed domestically.

For an MRIO analysis with M regions and N sectors in each region, the total output of industry i in region r (x_i^r) can be represented as follows:

$$x_i^r = \sum_{s=1}^M \sum_{j=1}^N a_{ij}^{rs} x_j^s + \sum_{s=1}^M y_i^{rs} \quad (2)$$

where a_{ij}^{rs} ($r, s=1,2, \dots, M; i=1,2, \dots, N$) is defined as the proportion of the input from sector i in region r to the production of one unit of output from sector j in region s ,

y_{ij}^{rs} ($r, s=1,2, \dots, M; i=1,2, \dots, N$) represents the finished goods by sector i produced in region r and consumed in region s . Equation (2) can also be expressed in matrix form as

$$\mathbf{X} = \mathbf{A}\mathbf{X} + \mathbf{Y} \quad (3)$$

and further transformed as follows:

$$\mathbf{X} = (\mathbf{I} - \mathbf{A})^{-1}\mathbf{Y} \quad (4)$$

where $(\mathbf{I} - \mathbf{A})^{-1}$ is the Leontief inverse matrix. Given the output in each region, the region- and sector-specific production related to the consumption activity of a given region r can be calculated as follows:

$$\mathbf{X}_{\text{com}}^r = (\mathbf{I} - \mathbf{A})^{-1}\mathbf{Y}^r \quad (5)$$

where \mathbf{Y}^r is a vector for the region r that includes the final consumption produced domestically (y^{rr}) and the final consumption imported from other regions ($y^{r,s}, s \neq r$), and $\mathbf{X}_{\text{com}}^r$ is a vector of the total production over multiple regions and multiple sectors related to the consumption in the region r . By incorporating a vector of the emission intensity (\mathbf{F}) representing the region- and sector-explicit emissions embodied by one unit of product derived from a production-based emission inventory, the consumption-based emission (\mathbf{E}^r) in the region r is constructed as follows:

$$\mathbf{E}^r = \mathbf{F}(\mathbf{I} - \mathbf{A})^{-1}\mathbf{Y}^r \quad (6)$$

The primary MRIO analysis is conducted in 2011 using the latest economic data from version 9 of the Global Trade Analysis Project (GTAP) (see **Supplementary Information** for details) ⁴⁰. Additionally, we conduct a time-series MRIO analysis to examine the historical trend of BC emissions embodied in global trade using the World Input-Output Database (WIOD) ⁴¹. The GTAP data set has a more detailed classification for countries/regions that could help us distinguish the contributions from Pakistan, Nepal, and other Asian regions, but it does not provide yearly economic data. Since we only use the WIOD data set to analyse the trends in the consumption-based emissions of several major regions (i.e., the USA, Europe, China and India), the differences between the two databases will not cause any inconsistency in our results. The production- and consumption-based emissions in different countries/regions are further combined with the source-receptor emission sensitivity derived from adjoint simulations to estimate the relative contributions of different regions to the BC in the HTP from production and consumption perspectives. To distinguish the effects of consumption-based emissions from that of climate variabilities on the historical trends, the source-receptor emission sensitivity is fixed in the year 2011 for both the primary and time-series analysis.

Figure3 | BC emissions embodied in global trade and BC radiative forcing over the HTP. **a**, Consumption-based interregional flows of BC emissions. “Source” on the left side refers to indirect BC emissions of one region as a consumer that drives the direct emission of BC from other regions. “Receptor” on the right side refers to one region as a producer of BC that is driven by other regions as consumers. Black dots indicate the net interregional flow of BC embodied in global trade as the “Receptor” minus the “Source”. **b**, Production- (top bar) and consumption-based (bottom bar) snow albedo forcing ($W m^{-2}$) of BC averaged over the HTP glaciers for different regions and sectors. The results for India, China, Pakistan, Nepal, Middle Asia, and the rest of South Asia are displayed according to the x-axis in the top half. The results for the USA, Europe, Russia, Africa, Southeast Asia, Central Asia, and all other regions are displayed according to the x-axis on the bottom half. Error bars denote uncertainty ranges related to inter-annual climate variabilities. Please refer to Supplementary Figure 9 for the estimation of BC direct radiative forcing at the top-of-atmosphere. See details in Supplementary Data 4 and 5.

Line 169

The authors refer to the potential induction of significant climate change through embodied BC emission. The topic of climate change was not discussed in the

preceding paragraph. Please clarify how this links in with what was said directly before. At the moment the statement appears a bit out of context.

Sorry for confusion. This sentence does not refer to the preceding paragraph. It actually refers to the following discussion about the responsibility for BC radiative forcing over the HTP glaciers in the next section. We have revised this sentence to avoid confusion (see P5, L162-164).

“...Generally, the global trade leads to a virtual transport pathway of BC from remote regions to the HTP glaciers. This process aggravates the BC pollution towards the HTP glaciers.”

Line 182

Use the term "half of the radiative forcing". In fact, "radiative forcings" in its plural form is used several times throughout the text and must be corrected.

We have corrected all similar mistakes throughout the manuscript. For example, in Page5:

“...More than half of the radiative forcing can be attributed to direct residential emissions (e.g., cooking and heating).”

[L174 Page 5]

“...Given that substantial territorial emissions are embodied in global trade that can be assigned to other regions, the radiative forcing contributed by India and China as final consumers are nearly 10 % lower than as direct emitters (Figure 3b and Supplementary Figure 11).”

[L179 Page 5]

Section starting in line 173

My understanding is that you have so far shown which primary emitters contribute to the BC that is present near the HTP glaciers. These primary emitters are mainly India and China, but also Nepal and Pakistan. You also discussed further up that due to the atmospheric pathways, emissions that originate from Europe and the US are also present in the HTP glaciers. In the previous paragraph you discussed the consumption-based BC emissions (and also income-based, see my previous comment). I would have expected that in this paragraph you present a concept of how all BC emissions that are present near the glaciers can be accounted for in a consumer-based fashion. But my understanding is that you are only discussing the consumption-based responsibility for emissions from originating from India and China? I think it would be more convincing to include all emitters that contribute to the BC emissions near the glaciers in your discussion about responsibility. Further, it may be worth structuring this section into production- and consumption-based emissions, rather than switching between the two concepts.

Thanks for your constructive suggestion! We have revised the manuscript to provide a thorough discussion that includes all BC contributors to the HTP glaciers. In the previous section, we discussed the atmospheric transport pathways of BC directly emitted from different regions to the HTP glaciers, which was from a production-based perspective. In this section, we further focused on the impacts of international trade in modulating the BC emissions among different countries/regions by the interregional exchange of commodities and consequently effects the BC burden over the HTP from a consumption-based perspective. Therefore, we prefer to compare the difference of the consumption-based emissions relative to the production-based emissions in our discussions rather than separately describing their results in order to give a clear indication of the effects of global trade. Please refer to the “*Aggravation of BC pollution over the HTP glaciers due to international trade*” section in the revised manuscript or our response to the previous comment for details.

Line 217

The authors state that the amount BC emissions embodied in international trade is "two times smaller" than that of CO₂. I do not understand this statement. Are the BC emission only half as much as the CO₂ emissions in terms of mass? Or a quarter as much? The statement "two times smaller" should be changed to "half as much" or "a quarter of...". Also, the same mass of BC emissions might have different effects than the same amount of CO₂. It was mentioned that BC emissions only remain in the atmosphere for several weeks. As a result, increased radiative forcing can only appear during that time. Only the continuous practice of emitting BC will maintain constantly increased radiative forcing, and therefore glacier loss. But it is argued that the reduction of BC emission is necessary to avoid the immediate loss of the HTP glaciers. Can you quantify how much a ton of short-lived BC emissions compares to a ton of CO₂ in terms contribution to glacier loss? And hence, what positive effects this would have on the preservation of the glaciers.

Thanks for your question! We realize that it is inappropriate to compare the amount of BC emissions embodied in international trade with that of CO₂ due to their different properties. We have deleted this sentence in Line 214 to avoid confusions.

As we have mentioned before, BC was regarded as the second most important anthropogenic climate forcer causing the fast melting of the HTP glaciers after CO₂. BC in the atmosphere can warm the air by absorbing more radiation, then warm the surface and melt the snow by air-surface heat transfer. It can also directly heat the surface and melt the snow by reducing surface albedo when it deposits on the snow and ice. After emitted from incomplete combustion, BC persists in the atmosphere for a few weeks then deposit on the ground. Although the lifetime of BC in the atmosphere is short relative to CO₂, it exerts persistent effects on the HTP because of the continually emissions from human activities. More importantly, BC deposited on the snow and ice would not be washed away during the snow melting (Xu et al.,

2012a). Therefore, it can persist in the snow-pack for a long time and accumulate to large concentrations that may significantly accelerate the melting of HTP glaciers (Xu et al., 2012a). Considering the different properties between BC and CO₂, their mechanisms associated with the glacier loss are quite different. Therefore, it is difficult to compare the response of the HTP glaciers to per unit emission reduction of BC with that of CO₂. Previous studies have indicated that the response of surface temperature to per unit radiative forcing from BC on snow and ice was two to four times larger than that from CO₂ (Qian et al., 2011; Stocker et al., 2013).

Here we emphasize the important of BC mitigation over the HTP glaciers because the HTP glaciers are important water suppliers that feed billions of people in South Asia, Southeast Asia and East Asia. The continually glacier loss would add considerably to drought-related water stress that threatens the water availability and food security for hundreds of millions of people at present and in the future (Immerzeel et al., 2010; Menon et al., 2010b; Pritchard, 2017b). These consequences may be irreversible. Our study reveals that global trade transfers BC emissions from remote regions to regions where the HTP glaciers are more vulnerable to and consequently lead to an aggravation of this problem. Therefore, we suggest that the mitigation of BC is urgently needed and countries that primarily import commodities should express caution when selecting their trading partners to mitigate corresponding climate effects.

In our revised manuscript, we have rephrased our statement in the introduction and discussion to illuminate the value of this study (see P2, L63-71 and P6, L223-L229).

“...Ample evidence shows that the increasing amount of BC being transported into the HTP plays an important role on the observed rapid glacial retreat therein comparable to the effects of greenhouse gases (GHGs)^{2,6,12}. The response of surface temperature to per unit radiative forcing from BC on snow and ice was assessed to be two to four times larger than that from CO₂¹³. Unlike well-distributed and long-lived GHGs, BC has a heterogeneous spatiotemporal distribution that is extremely sensitive to the emission sites because of a much shorter residence time in the atmosphere. Therefore, tracing the sources of BC being transported into the HTP is essential for providing valuable guidance for an effective mitigation strategy.”

[L63-71, Page2]

“The HTP glaciers, which constitute a crucial water resource, are suffering from rapid retreat due to global warming^{4,5}. Both BC and CO₂ represent major climate factors; however, while BC persists in the atmosphere for only a few weeks, CO₂ can remain in the air for centuries. After depositing on the snow and ice, BC directly warms the cryosphere and accelerates the snow-melting more efficiently than CO₂ owing to a positive albedo feedback^{13,32}. The mitigation of BC could lead to

significant climatic benefits and is therefore urgently needed to preserve the HTP glaciers³³.”

[L223-229, Page6]

Line 225

You mention a time-series here. Was this time series part of your discussion before?

Thanks for your question! The time-series analysis is one of the sparking points in this study. We analyzed the historical trends of consumption-based contributions to BC over the HTP glaciers from 2000 to 2014. It shows that the contribution from India to BC over the HTP glaciers has substantially increased over the last decade while the USA, Europe, and China all demonstrate a decreasing contribution trend. The India plays more and more important roles in global trade. Considering its significant effects on the BC pollutions over the HTP glaciers, the increasing contributions of India to the HTP may lead to severe effects over the HTP glaciers. In the revised manuscript, we have discussed this time series and its implications more thoroughly (see P6, L200-220).

“Trends of anthropogenic contributions to BC over the HTP glaciers

The historical trends of anthropogenic contributions to BC over the HTP glaciers from 2000 to 2014 are estimated from a consumption perspective (Figure 4). It indicates that the contribution from India to BC over the HTP glaciers has substantially increased over the last decade while the USA, Europe, and China all demonstrate a decreasing contribution trend. Specifically, the increasing contribution from India can be mainly attributed to the rapid growth of industrial sectors, especially the transportation and construction. The significant decrease over China was primarily induced by the mitigation of residential emissions over the last decade. The industrial activities over China have increasingly threatened the HTP glaciers from 2000 to 2014 (Figure 4). Recently, China has implemented a series of clean air actions to reduce emissions of atmospheric pollutants and hence the industrial emission of BC in China has begun to decrease since 2014²⁹. Meng, et al.³⁰ showed that some labor-intensive and emission-intensive production activities are moving from China to other developing countries including India because of rising labor costs and industry structure change. Evidence has shown that the shifting of polluting industries to regions with more permissive environmental regulations has become a substantial and growing problem³¹. Considering that the HTP glaciers are more vulnerable to BC emitted from India than that from China, we infer that the rise of trade among developing countries and the increasing emissions from India may enhance BC pollutions over the HTP glaciers.”

Line 235

Delete the "the" from "the global trade".

We have revised it (P7, L239).

“Furthermore, our study reveals that global trade lead to a virtual transport of BC from remote regions to the HTP glaciers and that this virtual pathway is much more efficient than the typical atmospheric transport pathway²⁵.”

Line 241

If the HTP is more affected by emissions from adjacent regions, it may not have been necessary to assess the atmospheric pathways of BC emission from other countries in the first place?

Our study reveals that global trade transfers BC emissions from remote regions to regions where the HTP glaciers are more vulnerable to and consequently aggravates the BC pollutions over the HTP. Hence, we suggest that countries should care more about the selection of their trading partners in order to minimize the corresponding climate effects. The assessment of the atmospheric pathways of BC provides us a detailed and quantitative source-receptor sensitivity map of BC in the HTP. For example, China and India are two important BC source regions surrounding the HTP. Our analysis shows that although the amount of BC emitted from China is much larger than the amount of BC emitted from India, the contributions of BC from China to the HTP is only half of that from India due to the mid-latitude westerlies. It means that the sensitivities of HTP to different regions are quite different, even if these regions are adjacent regions. These sensitivities may be more complex due to global trade. For example, Meng et al. (2018) showed that production activities are continually relocating from China to other developing countries including India in the last decades. Therefore, it is valuable for us to quantify the contributions from different countries/sectors as direct emitters and final consumers in global trade to the BC pollution over the HTP glaciers and the historical trends. We have revised this sentence to clarify our statement in the revised manuscript (P8, L243-246).

“Since the sensitivities of BC over the HTP glaciers to various source regions show large differences, our study suggests that countries that primarily import commodities should care more about the selection of their trading partners to mitigate corresponding climate effects.”

Line 269

The statement "Due to the coarse resolution of the adjoint model, it is difficult to directly attribute the BC being transported to the HTP to a particular country" gives the impression that the foundation for your previous discussion is shaky. If it is not possible to reliably account the BC emissions above the HTP to a certain country, then it would be hard to defend your results. I would suggest you discuss the

assumptions you had to make for your study in detail before you commence your discussion.

Thank you for your suggestion. We meant that the grid in adjoint model cannot match the territorial boarder perfectly, which may introduce some uncertainty. We have followed methods in previous studies (Xu et al., 2018; Zhang et al., 2009) to reduce uncertainty, and the robustness of such multi-model studies has been proved by Lin et al. (2016) and Zhang et al. (2017). Previous studies interested in the source attributions of BC over the HTP have used similar resolutions in their model simulations (Kopacz et al., 2011; Lu et al., 2012; Zhang et al., 2015). Model evaluations shows that our model exhibits a good performance in the simulation of the atmospheric transport and deposition of BC over the HTP, providing us the confidence to use this model framework to analysis BC source–receptor relationships in the HTP.

Considering that the long-distance atmospheric transport of BC to the HTP glaciers are following the large-scale circulations in the atmosphere (e.g., the mid-latitude westerlies), the $2^{\circ} \times 2.5^{\circ}$ resolution is sufficient for us to quantify the long-distance transport of BC from other regions to the HTP. Here we downscaled the $2^{\circ} \times 2.5^{\circ}$ source-receptor sensitivity map of BC in the HTP proportionally to a smaller resolution map based on the high-resolution emission data set. The PKU-BC-Inventory provides us a $0.1^{\circ} \times 0.1^{\circ}$ emission inventory of BC. The contribution of a $2^{\circ} \times 2.5^{\circ}$ model grid derived from the adjoint analysis was further assigned to a set of $0.1^{\circ} \times 0.1^{\circ}$ grids by ratios according to their emission intensities. Similar approaches have been widely used in previous studies(Xu et al., 2018; Zhang et al., 2009). Here it was based on a basic principle that regions emitted more BC should share more responsibilities. This approach provides great beneficial results for our estimation. We have clarified our statement in the revised manuscript (see P8, L274-280). The relevant uncertainties have been discussed in the **Supplementary Information** (see P7, L281-289).

In the main text:

“...The $2^{\circ} \times 2.5^{\circ}$ source-receptor sensitivity map of BC in the HTP derived from the adjoint analysis is disaggregated to $0.1^{\circ} \times 0.1^{\circ}$ using high-resolution emission inventory as a proxy. The emission intensity at a $0.1^{\circ} \times 0.1^{\circ}$ resolution is obtained from the PKU-BC-Inventory³⁹ (Supplementary Figure 2). This approach is based on a basic principle that regions emitted more BC may have more contributions to the HTP and should share more responsibilities. It considers the sub-grid distribution of BC emissions and hence covers the shortage of the model resolution more or less.”

[L274-280, Page 8]

In the **Supplementary Information**:

“...In this study, we downscaled the $2^{\circ} \times 2.5^{\circ}$ source-receptor sensitivity map of BC over the HTP to a finer resolution using emission as a proxy, which assumes that the transport efficiency is the same within a model grid. Considering that the long-

distance atmospheric transport of BC to the HTP glaciers are following the large-scale circulations in the atmosphere, the approach has negligible effects on the estimation of most countries/regions. As for the adjacent regions like Nepal, which are beneath the HTP, the sub-grid transport efficiency may be different due to the complex terrain effects. Hence, it may contribute to the uncertainties in estimations of these regions.”

[L281-289, Page 7]

Line 293

Are the atmospheric pathways invariant for different years? Or do different years have different weather patterns that result in BC emission originating from different regions? If yes, this must be considered when running the economic analysis for different years.

Thanks for your questions! We agree that the variabilities of climate conditions can affect the inter-annual variabilities of the atmospheric pathways. We have conducted a multiyear simulation from 2007 to 2011 with varying climate conditions and fixed emissions to estimate the inter-annual variabilities in the climate and the corresponding changes in atmospheric transport pathways. It shows that the uncertainties of our estimation related to the inter-annual climate variabilities are generally small. Since our analysis is focused on the long-term trends instead of the inter-annual variations, the variabilities of climate conditions exert negligible effects on our analysis. Here we use the same climate condition for different years in our analysis to distinguish the effects of consumption-based emissions from that of climate variabilities on the historical trends. We have revised the **Materials and Methods** section to make it clear (P8, L280-L287 and P9, L325-L339).

“...The adjoint simulations are conducted for all months from 2007 to 2011 with BC emissions fixed in 2011. Each case spans three weeks of the corresponding month and the model response (Y) is defined for the last week. Our analysis focuses on the results of 2011 to ensure consistency with the multi-region input-output (MRIO) analysis. Meanwhile, we use the multiyear result to estimate the uncertainty range associated with the inter-annual variabilities in the climate and the corresponding changes in atmospheric transport pathways.”

[L280-287, Page 8]

“...Additionally, we conduct a time-series MRIO analysis to examine the historical trend of BC emissions embodied in global trade using the World Input-Output Database (WIOD) ⁴¹. ...The production- and consumption-based emissions in different countries/regions are further combined with the source-receptor emission sensitivity derived from adjoint simulations to estimate the relative contributions of different regions to the BC in the HTP from production and consumption perspectives. To distinguish the effects of consumption-based emissions from that of climate variabilities on the historical trends, the source-receptor emission sensitivity is fixed in the year

2011 for both the primary and time-series analysis.”

[L325-339, Page 9]

Line 303

I believe that mixing the two databases could lead to inconsistencies. Were you able to settle on one database for the emissions data that you used? Also, the Eora database offers both full geographical and temporal data coverage for your study. But it does not have the same resolution as GTAP for Pakistan or Nepal.

In this study, we used the GTAP database to estimate BC emissions from different regions/sectors embodied by the global trade. The GTAP data set has a more detailed classification for countries/regions that could help us distinguish the contributions from Pakistan, Nepal, and other Asian regions, but it does not provide yearly economic data. Therefore, we further used the WIOD database to analysis the historical trends in the consumption-based emissions of several major regions (i.e., the USA, Europe, China and India). We understand the reviewer’s concern for the two databases. However, since these two databases are used for different purpose, we think that the differences between them will not cause any conflicts in our results. We have revised our statement to make it clear (P9, L327-L333).

“...The GTAP data set has a more detailed classification for countries/regions that could help us distinguish the contributions from Pakistan, Nepal, and other Asian regions, but it does not provide yearly economic data. Since we only use the WIOD data set to analyse the trends in the consumption-based emissions of several major regions (i.e., the USA, Europe, China and India), the differences between the two databases will not cause any inconsistency in our results.”

Figure 4, arrows

Does Figure 4 show the consumption-based effects, or the production-based effects?
Do the arrows point towards the areas in the HTP where the effects occur?

Figure 4 shows historical trends of consumption-based anthropogenic contributions to BC over the HTP glaciers. The arrows therein generally indicated the effects of four different regions on the HTP glacier regions. Their thicknesses represent the magnitudes of relevant contributions. It provided a general view about the virtual transport pathways of BC from remote regions to the HTP glaciers embodied in global trade. We have modified these arrows in Figure 4.

Figure 4 | Historical trends of consumption-based anthropogenic contributions to BC over the HTP glaciers. Annual-mean contributions (10^5 kg) from four different regions (i.e., the USA, Europe, China, and India) and 13 sectors to the BC mass-burden over the HTP glacier regions from consumption perspective during the period from 2000 to 2014. The thickness of an arrow represents the magnitude of the contribution. See details of in Supplementary Data 6.

Supplementary Figure 3

I think that a similar figure to this one for BC emissions in the HTP region only would be greatly beneficial for the main text.

Thanks for your suggestion! We have added a figure in our revised manuscript for the distributions of direct geographic BC emissions, which provides a detailed view for BC emissions in the HTP region (please refer to Supplementary Figure 2 below).

Supplementary Figure 2 | Distributions of BC anthropogenic emission in 2011. The BC emission data set with a $0.1^\circ \times 0.1^\circ$ resolution is obtained from the PKU-BC-Inventory¹⁷.

References:

- Bauer, S.E., Bausch, A., Nazarenko, L., Tsigaridis, K., Xu, B., Edwards, R., Bisiaux, M., McConnell, J., 2013. Historical and future black carbon deposition on the three ice caps: Ice core measurements and model simulations from 1850 to 2100. *Journal of Geophysical Research: Atmospheres* 118, 7948-7961.
- Bond, T.C., Doherty, S.J., Fahey, D., Forster, P., Berntsen, T., DeAngelo, B., Flanner, M., Ghan, S., Kärcher, B., Koch, D., 2013. Bounding the role of black carbon in the climate system: A scientific assessment. *Journal of Geophysical Research: Atmospheres* 118, 5380-5552.
- Immerzeel, W.W., Van Beek, L.P., Bierkens, M.F., 2010. Climate change will affect the Asian water towers. *Science* 328, 1382-1385.
- Kopacz, M., Mauzerall, D.L., Wang, J., Leibensperger, E.M., Henze, D.K., Singh, K., 2011. Origin and radiative forcing of black carbon transported to the Himalayas and Tibetan Plateau. *Atmospheric Chemistry and Physics* 11, 2837-2852.
- Kraaijenbrink, P.D.A., Bierkens, M.F.P., Lutz, A.F., Immerzeel, W.W., 2017. Impact of a global temperature rise of 1.5 degrees Celsius on Asia's glaciers. *Nature* 549, 257-260.
- Liang, S., Qu, S., Zhu, Z., Guan, D., Xu, M., 2017. Income-Based Greenhouse Gas Emissions of Nations. *Environmental Science & Technology* 51, 346-355.
- Lin, J., Tong, D., Davis, S., Ni, R., Tan, X., Pan, D., Zhao, H., Lu, Z., Streets, D., Feng, T., 2016. Global climate forcing of aerosols embodied in international trade. *Nature Geoscience* 9, 790-794.
- Lu, Z., Streets, D.G., Zhang, Q., Wang, S., 2012. A novel back-trajectory analysis of the origin of black carbon transported to the Himalayas and Tibetan Plateau during 1996-2010. *Geophysical Research Letters* 39, n/a-n/a.
- Meng, J., Mi, Z.F., Guan, D.B., Li, J.S., Tao, S., Li, Y., Feng, K.S., Liu, J.F., Liu, Z., Wang, X.J., Zhang, Q., Davis, S.J., 2018. The rise of South-South trade and its effect on global CO₂ emissions. *Nature Communications* 9.
- Menon, S., Koch, D., Beig, G., Sahu, S., Fasullo, J., Orlikowski, D., 2010a. Black carbon aerosols and the third polar ice cap. *Atmos. Chem. Phys.* 10, 4559-4571.
- Menon, S., Koch, D., Beig, G., Sahu, S., Fasullo, J., Orlikowski, D., 2010b. Black carbon aerosols and the third polar ice cap. *Atmospheric Chemistry and Physics* 10, 4559-4571.
- Pritchard, H.D., 2017a. Asia's glaciers are a regionally important buffer against drought. *Nature* 545, 169-174.
- Pritchard, H.D., 2017b. Asia's glaciers are a regionally important buffer against drought. *Nature* 545, 169.
- Qian, Y., Flanner, M.G., Leung, L.R., Wang, W., 2011. Sensitivity studies on the impacts of Tibetan Plateau snowpack pollution on the Asian hydrological cycle and monsoon climate. *Atmospheric Chemistry and Physics* 11, 1929-1948.
- Qian, Y., Yasunari, T.J., Doherty, S.J., Flanner, M.G., Lau, W.K.M., Ming, J., Wang, H., Wang, M., Warren, S.G., Zhang, R.J.A.i.A.S., 2015. Light-absorbing particles in snow and ice: Measurement and modeling of climatic and hydrological impact. *Journal of Geophysical Research: Atmospheres* 120, 64-91.
- Ramanathan, V., Chung, C., Kim, D., Bettge, T., Buja, L., Kiehl, J.T., Washington, W.M., Fu, Q., Sikka, D.R., Wild, M., 2005. Atmospheric brown clouds: Impacts on South

Asian climate and hydrological cycle. *Proceedings of the National Academy of Sciences of the United States of America* 102, 5326-5333.

Ramanathan, V., Ramana, M.V., Roberts, G., Kim, D., Corrigan, C., Chung, C., Winker, D., 2007. Warming trends in Asia amplified by brown cloud solar absorption. *Nature* 448, 575-U575.

Steininger, K.W., Lininger, C., Meyer, L.H., Munoz, P., Schinko, T., 2016. Multiple carbon accounting to support just and effective climate policies. *Nature Climate Change* 6, 35-41.

Stocker, T., Qin, D., Plattner, G., Tignor, M., Allen, S., Boschung, J., Nauels, A., Xia, Y., Bex, B., Midgley, B., 2013. IPCC, 2013: climate change 2013: the physical science basis. Contribution of working group I to the fifth assessment report of the intergovernmental panel on climate change. Cambridge University Press.

Wang, R., Tao, S., Balkanski, Y., Ciais, P., Boucher, O., Liu, J., Piao, S., Shen, H., Vuolo, M.R., Valari, M., 2014. Exposure to ambient black carbon derived from a unique inventory and high-resolution model. *Proceedings of the National Academy of Sciences* 111, 2459-2463.

Wang, R., Tao, S., Wang, W., Liu, J., Shen, H., Shen, G., Wang, B., Liu, X., Li, W., Huang, Y., 2012. Black carbon emissions in China from 1949 to 2050. *Environmental science & technology* 46, 7595-7603.

Xu, B., Cao, J., Joswiak, D.R., Liu, X., Zhao, H., He, J., 2012a. Post-depositional enrichment of black soot in snow-pack and accelerated melting of Tibetan glaciers. *Environmental Research Letters* 7, 014022.

Xu, B.Q., Cao, J.J., Joswiak, D.R., Liu, X.Q., Zhao, H.B., He, J.Q., 2012b. Post-depositional enrichment of black soot in snow-pack and accelerated melting of Tibetan glaciers. *Environmental Research Letters* 7.

Xu, Y., Shen, H., Yun, X., Gao, F., Chen, Y., Li, B., Liu, J., Ma, J., Wang, X., Liu, X., Tian, C., Xing, B., Tao, S., 2018. Health effects of banning beehive coke ovens and implementation of the ban in China. 115, 2693-2698.

Yasunari, T.J., Bonasoni, P., Laj, P., Fujita, K., Vuillermoz, E., Marinoni, A., Cristofanelli, P., Duchi, R., Tartari, G., Lau, K.M., 2010. Estimated impact of black carbon deposition during pre-monsoon season from Nepal Climate Observatory – Pyramid data and snow albedo changes over Himalayan glaciers. *Atmos. Chem. Phys.* 10, 6603-6615.

Zhang, Q., Jiang, X., Tong, D., Davis, S.J., Zhao, H., Geng, G., Feng, T., Zheng, B., Lu, Z., Streets, D.G., 2017. Transboundary health impacts of transported global air pollution and international trade. *Nature* 543, 705-709.

Zhang, R., Wang, H., Qian, Y., Rasch, P.J., Easter, R.C., Ma, P.L., Singh, B., Huang, J., Fu, Q., 2015. Quantifying sources, transport, deposition, and radiative forcing of black carbon over the Himalayas and Tibetan Plateau. *Atmospheric Chemistry and Physics* 15, 6205-6223.

Zhang, Y., Tao, S., Shen, H., Ma, J., 2009. Inhalation exposure to ambient polycyclic aromatic hydrocarbons and lung cancer risk of Chinese population. 106, 21063-21067.

Response to Anonymous Reviewer 3:

(Note: Reviewer comments are listed in blue, and responses to reviewer comments are in black. Pasted text from the new version of the paper is in italics.)

Summary - This paper calculates forcing by black carbon emitted from different kinds of trade and different producers and consumers. To do this, the authors connected an input-output model with an atmospheric model that simulates transport and deposition of black carbon on the glaciers. Then each of the estimated forcings is attributed to its cause, i.e. internal activity, or international trade.

Reaction – The models used for the analysis are well accepted and so the result appears valid. However, the presentation doesn't provide new insights and yet it is sensationalized and that leads to a negative impression.

It is known that modeled black carbon forcing is large over Himalayas. It is known that especially China has a big role in trade and some papers on pollutant attributed to trade have been published, including the authors' own work. This paper does add quantitatively to the estimates. But for many countries all the methods (Production, Consumption, Income) of looking at the contribution to BC forcing are similar. This is because many of the sources are internal to countries and not related to trade as the authors point out. They also point out “the percentage of global BC emissions embodied in the global trade is nearly two times smaller than that of CO₂” so CO₂ is embedded in trade more than BC. Why then is the paper title emphasizing trade? Perhaps it should emphasize the internal activities, instead!

Then, the title again states there is a “huge climate forcing.” This forcing isn't greater than that shown in other papers. And it could not be as the model is not new. What is huge? Is it bigger than other forcings? Again the science appears supportable, but the dramatic statement, and uncritical policy statements like “regions... should be cautious about their direct emissions” suggest a lack of sophistication.

We thanks the reviewer's comments. We agree that the BC radiative forcing over the HTP from the production perspective and the effects of international trade in modulating the global BC emissions have been documented in previous studies, however, separately. A holistic view on the BC pollutions over the HTP along the entire supply chain, involving all the partners is still absent. Given the critical roles of HTP in regulating the Asian hydrological cycle and climate system and the observed rapid glacier retreat in the HTP, it is necessary for us to study this issue from a cross-disciplinary perspective. The novelty and significance of this study can be reflected in the following aspects:

(1) A quantitative analysis of effects embodied in global trade on BC pollutions over

the HTP is strongly necessary. We agree with the reviewer that internal activities contribute to a large part of BC emissions for many countries. The domestic contributions to the BC pollutions over the HTP have raised much attention in previous studies (Kopacz et al., 2011; Li et al., 2016; Lu et al., 2012; Zhang et al., 2015). These results indicated that India should take responsibility for glaciers melting in the HTP induced by anthropogenic BC, however, ignored the potential effects of global trade in transferring BC emission from USA and Europe. Our study focused on the global trade effect is a valuable complementarity for previous studies. The property of BC is quite different from that of CO₂. CO₂ is nearly uniformly distributed in the atmosphere after emitted from human activities due to its chemical inertness. The amount of CO₂ over the HTP relies on the total amount of CO₂ emitted from global-wide rather than the exact locations of these emissions. It means that although global trade exerts significant effects on transferring the CO₂ emissions among different regions, its corresponding effects on the HTP glaciers can be greatly diluted. BC, in the contrary, has a heterogeneous spatiotemporal distribution due to its relative short lifetime in the atmosphere. BC over the HTP is extremely sensitive to the locations of emission sources. The interregional BC flow embodied in global trade can directly changes the BC burden over the HTP and cause different climate effects. Though the percentage of global BC emissions embodied in the global trade is smaller than that of CO₂, its potential effects on the HTP may be more significant. These effects cannot be directly estimated based on previous knowledge without a quantitative analysis.

- (2) We for the first time quantify the contributions from different countries/sectors to the BC pollution over the HTP glaciers from both production and consumption perspectives by integrating multi-models. The adjoint inverse model, different from the traditional chemistry transport model (CTM), provides a source-receptor sensitivity quantification for BC over the HTP in a model grid level. It traces the BC pollution over the HTP backward to regions where the direct geographic emission occurs. The multi-region input-output (MRIO) analysis further traces the flow of direct geographic emissions to regions where the related goods and services are ultimately consumed. The combination of the adjoint model and the MRIO model can therefore assess the contributions of a specific country/region to BC pollutions over the HTP from various perspectives and identify the essential driving factors of these pollutions. This approach can provide valuable information for policy-makers to get an effective mitigation strategy.
- (3) The effect of global trade on the BC pollutions over the HTP is large enough to draw public attention. In our analysis, we find that global trade transfer BC emissions from remote regions like US and Europe to Asian regions where the HTP is more vulnerable to. Regions like USA and Europe may exact nearly ten times larger effects on the HTP in global trade than their direct contributions though atmospheric transport pathways. Our estimation shows that nearly 10 % of BC over the HTP glaciers can be attribute to the effects of global trade though the relocation

of production activities and exchange of commodities. Additionally, our analysis reveals that the contribution from India to BC over the HTP glaciers may continually increase partly due to the rise of trade among developing countries. As one of the most important factor causing the fast melting of HTP glaciers, studies showed that nearly half of the Himalayan glaciers warming from 1950 to present is caused by BC aerosols (Qian et al., 2015; Ramanathan et al., 2005; Ramanathan et al., 2007). The response of surface temperature to per unit radiative forcing from BC on snow and ice was assessed to be two to four times larger than that from CO₂ (Qian et al., 2011; Stocker et al., 2013). Considering the severe BC pollutions over the HTP glaciers, we believe that the effect of global trade is large enough for us to be concerned.

In our manuscript, the title has been revised as “***How Do Global Trade Cascade to Large Climate Forcing over the Tibetan Plateau Glaciers***”. We have changed the expression from “huge” to “large” to avoid the overstatement. We have expanded the introduction and discussion to illuminate the value of this study (see P2, L63-71 and P6, L223-L229). We have also provided a more thoroughly discussion about the effects of global trade on the BC pollution over the HTP glaciers (e.g., P5, L159-164 and L191-194). Relevant sentences have been rephrased to avoid the uncritical policy statement (e.g., P7, L235-237). Additionally, we have provided more details about the methods used in this study and added more updated observations for model evaluation to show the reliable foundation of our analysis. Our manuscript has been significantly improved by addressing these issues. Please see our response to each comment below and refer to the revised manuscript for more details.

“...Ample evidence shows that the increasing amount of BC being transported into the HTP plays an important role on the observed rapid glacial retreat therein comparable to the effects of greenhouse gases (GHGs)^{2,6,12}. The response of surface temperature to per unit radiative forcing from BC on snow and ice was assessed to be two to four times larger than that from CO₂¹³. Unlike well-distributed and long-lived GHGs, BC has a heterogeneous spatiotemporal distribution that is extremely sensitive to the emission sites because of a much shorter residence time in the atmosphere. Therefore, tracing the sources of BC being transported into the HTP is essential for providing valuable guidance for an effective mitigation strategy.”

[L63-71, Page 2]

“...Our estimation shows that nearly 10 % of the BC over the HTP glaciers are embodied by interregional flow within global trade though the importation of commodities. Generally, the global trade leads to a virtual transport pathway of BC from remote regions to the HTP glaciers. This process aggravates the BC pollution towards the HTP glaciers.”

[L159-164, Page 5]

“...Furthermore, the USA and Europe overall contribute BC SAF of 0.11 W m⁻² to the HTP glaciers as final consumers (Figure 3b). These effects are nearly ten times larger than their direct contributions though atmospheric transport pathways.”

[L191-194, Page 5]

“...The HTP glaciers, which constitute a crucial water resource, are suffering from rapid retreat due to global warming^{4,5}. Both BC and CO₂ represent major climate factors; however, while BC persists in the atmosphere for only a few weeks, CO₂ can remain in the air for centuries. After depositing on the snow and ice, BC directly warms the cryosphere and accelerates the snow-melting more efficiently than CO₂ owing to a positive albedo feedback^{13,32}. The mitigation of BC could lead to significant climatic benefits and is therefore urgently needed to preserve the HTP glaciers³³.”

[L223-229, Page 6]

“...Other Asian countries/regions, especially Pakistan and Nepal, should also mitigate their anthropogenic emissions due to their high source-receptor sensitivities to the HTP glaciers.”

[L235-237, Page 7]

Scientific comments: (Addressing any one of these could have made the analysis more novel.)

1. The transport model used here is coarse resolution and has difficulty representing transport to Himalayas.

Thanks for your question. In this study, we use the GEOS-Chem model and its adjoint to analysis the origin of BC transported to the HTP. This model has been widely used in previous studies to simulate the BC transport. For example, He et al. (2014b) has provided a systematically evaluation for the BC simulations over the HTP. In this study, we further evaluate its performance using updated in situ observations (see Supplementary Table 4 and Supplementary Table 6). These results all show that the GEOS-Chem model exhibits a good performance in the simulation of the atmospheric transport and deposition of BC over the HTP. Therefore, we think that the GEOS-Chem model is reliable for this analysis. We agree that the relatively coarse resolution of global chemistry transport model (CTM) restricted its performance in simulating the complex topography and meteorological field within the HTP, which is an inherent problem existed in most of chemistry transport models. However, our analysis is focused on the long-distance transport of BC from other regions to the HTP following the large-scale circulations in the atmosphere. The transport efficiency associated with the large-scale circulations has negligible differences within a model grid for most regions. Our estimations are in consistent with previous studies interested in the HTP (Kopacz et al., 2011; Lu et al., 2012; Zhang et al., 2015). The effects related to the coarse resolution are mainly exerted on the adjacent regions beneath the HTP (e.g., Nepal) where the sub-grid transport efficiency may be different over the complex

terrains. In this study, we downscaled the $2^\circ \times 2.5^\circ$ model results to $0.1^\circ \times 0.1^\circ$ using emission as a proxy following previous studies (Xu et al., 2018; Zhang et al., 2009b). It is based on a basic principle that regions emitted more BC may have more contributions to the HTP and should share more responsibilities. Although this approach does not directly consider the complex terrain effects, which has not been well understood yet, it considers the sub-grid distribution of BC emissions and covers the shortage of the model resolution more or less.

We have provided more details about the down-scaling approach in the revised manuscript (see P8, L274-280). The relevant uncertainties have been discussed in the **Supplementary Information** (see P7, L281-289).

In the main text:

“...The $2^\circ \times 2.5^\circ$ source-receptor sensitivity map of BC in the HTP derived from the adjoint analysis is disaggregated to $0.1^\circ \times 0.1^\circ$ using high-resolution emission inventory as a proxy. The emission intensity at a $0.1^\circ \times 0.1^\circ$ resolution is obtained from the PKU-BC-Inventory³⁹ (Supplementary Figure 2). This approach is based on a basic principle that regions emitted more BC may have more contributions to the HTP and should share more responsibilities. It considers the sub-grid distribution of BC emissions and hence covers the shortage of the model resolution more or less.”

[L274-280, Page 8]

In the **Supplementary Information**:

“...In this study, we downscaled the $2^\circ \times 2.5^\circ$ source-receptor sensitivity map of BC over the HTP to a finer resolution using emission as a proxy, which assumes that the transport efficiency is the same within a model grid. Considering that the long-distance atmospheric transport of BC to the HTP glaciers are following the large-scale circulations in the atmosphere, the approach has negligible effects on the estimation of most countries/regions. As for the adjacent regions like Nepal, which are beneath the HTP, the sub-grid transport efficiency may be different due to the complex terrain effects. Hence, it may contribute to the uncertainties in estimations of these regions.”

[L281-279, Page 7]

Supplementary Table 4 | Observed and simulated surface BC concentrations over the Himalayas and the Tibetan Plateau (HTP) and adjacent regions.

Site	Time	BC in surface air ($\mu\text{g m}^{-3}$)					
		Obs.*	Exp.A	Exp.B	Exp.C	Exp.D	Exp.E
Nainital	2006 Jan-May	1.4 ^a	0.50	0.49	0.71	0.93	0.92
Kharagpur	2005-2007	6.2±4 ^b	1.42	1.37	2.43	2.90	2.85
Kanpur	2007-2008	3.8±2.3 ^c	0.99	0.96	1.71	1.68	1.56
Gandhi College	Sep.-Dec. 2006	4.75±0.48 ^d	1.78	1.75	2.58	2.82	2.71
Nagarkot	1999-2000	1.0±0.63 ^e	0.35	0.31	0.43	0.69	0.64

Langtang	1999-2000	0.38±0.34 ^e	0.32	0.29	0.39	0.65	0.60
NCOP	2006-2008	0.15±0.14 ^f	0.24	0.22	0.36	0.80	0.77
Manora Peak	2005-2008	1.14±0.55 ^g	0.35	0.33	0.46	0.62	0.59
NCOS	2006	0.08±0.07 ^h	0.09	0.08	0.14	0.23	0.21
	2012	0.19±0.11 ⁱ	0.09	0.08	0.14	0.23	0.21
Zhuzhang	Aug.2004-Feb.2005	0.34±0.18 ^j	0.13	0.11	0.26	0.33	0.29
Muztagh Ata	2003-2006	0.06±0.04 ^k	0.15	0.05	0.06	0.27	0.27
Hanle	2009-2010	0.08±0.06 ^l	0.06	0.04	0.05	0.08	0.07
Lulang	2008-2009	0.52±0.35 ^m	0.06	0.03	0.05	0.14	0.11
QOMS	Aug. 2009-Jul. 2010	0.25±0.2 ⁿ	0.20	0.17	0.29	0.73	0.71
	May 2015-May 2017	0.3±0.34 ^o	0.20	0.17	0.29	0.73	0.71
Beiluhe	Nov. 2012-Jun. 2013	0.4 ^p	0.05	0.04	0.07	0.11	0.10
Ranwu	Nov. 2012-Jun. 2013	0.14 ^p	0.12	0.10	0.18	0.30	0.26
QSSGEE	May 2009-Mar. 2011	0.1 ^q	0.09	0.08	0.16	0.28	0.24

* Multi-month averaged values and standard deviations are provided for different measurement sites. Sources: ^aBeegum et al.(2009), ^bNair et al. (2012), ^cRam et al. (2010b), ^dGanguly et al. (2009), ^eCarrico et al. (2003), ^fBonasoni et al. (2010), ^gRam et al. (2010a), ^hMing et al. (2010), ⁱWan et al.(2015), ^jQu et al. (2008), ^kCao et al. (2009), ^lBabu et al. (2011), ^mZhao et al. (2013), ⁿCong et al.(2015), ^oChen et al.(2018), ^pWang et al.(2016), ^qZhao et al.(2012).

Supplementary Table 6 | Observed and simulated BC concentrations in snow over the HTP and adjacent regions

Site	Time	BC in snow ($\mu\text{g kg}^{-1}$)					
		Obs.	Exp.A	Exp.B	Exp.C	Exp.D	Exp.E
Zuoqiupu	Monsoon 2006 ^a	7.9	21.01	11.42	30.43	38.08	19.53
	Non-monsoon 2006 ^a	15.9	42.81	37.64	49.67	76.84	48.77
Qiangyong	Summer 2001 ^b	43.1	27.18	23.49	41.23	26.30	18.77
Noijin Kangsang	Annual 2005 ^a	30.6	41.05	26.15	52.65	31.58	24.49
	Monsoon 2001 ^c	35	30.64	30.72	24.15	25.32	17.10
	Non-monsoon 2001 ^c	21	60.03	57.09	28.85	43.78	26.33
East Rongbuk	Summer 2002 ^c	20.3	30.31	30.96	24.40	24.08	16.41
	Oct.2004 ^d	18	39.27	38.98	22.33	24.15	26.00
	Sept.2006 ^e	9	31.62	30.02	23.38	29.05	19.17
	May 2007 ^f	41.8	35.22	36.79	25.94	21.45	17.61
Kangwure	Summer 2001 ^b	21.8	20.16	15.24	21.45	22.66	13.88
Namunani	Summer 2004 ^b	4.3	10.72	10.72	10.72	10.72	10.72
Mt.Muztagh	Summer 2001 ^b	37.2	25.60	23.54	27.54	43.52	36.70
	1999 ^b	26.6	22.23	17.59	38.59	76.92	47.82
Laohugou #12	Oct.2005 ^d	35	34.88	35.49	50.25	31.63	55.24
Qiyi	Jul.2005 ^g	22	35.54	24.26	35.23	49.07	31.95
1 July glacier	Summer 2001 ^b	52.6	25.98	19.87	24.91	30.06	24.31

Meikuang	Nov.2005 ^h	81	11.83	8.97	15.64	29.78	22.13
	Summer 2001 ^b	18.2	12.09	5.23	16.91	21.28	12.34
Dongkemadi	2005 ^e	36	12.96	6.58	14.50	25.05	16.83
	Aug.2014-Oct.2015 ⁱ	41.8	11.17	5.89	15.48	20.33	13.33
La'nong	Jun.2005 ^e	67	16.45	7.61	27.25	19.43	19.17
	Jul.2006 ^g	87.4	14.67	6.18	23.98	31.72	16.09
Zhadang	Jul.-Aug. 2012 ^j	91.8	16.60	6.80	24.57	28.55	16.02
Haxilegen River	Oct.2006 ^k	46.9	52.85	50.80	74.72	39.73	119.66
Urumqi Riverhead	Nov.2006 ^h	141	39.13	38.22	98.20	51.20	146.67
Miao'ergou #3	Aug.2005 ^d	111	63.05	47.37	86.59	44.48	80.77
Demula glacier	Jun.2015 ^l	56.6±26.1	15.83	7.80	24.93	31.64	16.94
Muji glacier	Jul.2012 ^m	25	63.23	58.32	36.98	56.30	34.46
Tianshan Urumpi glacier #1	Aug. 2013 ⁿ	42.5	66.01	62.10	95.09	47.12	98.48

Sources: ^aXu et al. (2009), ^bXu et al.(2006),^cMing et al.(2008), ^dMing et al. (2009a), ^eMing et al. (2013), ^fMing et al. (2012), ^gMing et al. (2010), ^hMing et al. (2009b), ⁱLi et al.(2017), ^jQu et al.(2014), ^kMing et al.(2011a),^lZhang et al.(2016), ^mYang et al.(2015), ⁿMing et al.(2016).

2. Black carbon forcing is caused by change in snow albedo, snow albedo depends on dust and even little dust can reduce the forcing of black carbon. The dust treatment isn't discussed.

In this study, we used a stochastic snow albedo model (SSAM) to calculate the BC-snow albedo forcing (SAF). This model explicitly simulates multiple BC particles both stochastically and internally/externally mixed with various shapes of snow grains, and the mean value is used in the final report. The underlying ground albedo in the model was obtained from GEOS-5 meteorological fields. The BC-induced snow albedo reduction was computed via the adding/doubling radiative transfer method using the single scattering properties of the BC-snow mixtures. We agree that dust can lower snow albedo and hence affect the BC forcing. The IPCC AR5 reported that the mixture of mineral dust in the snow may reduce BC forcing by approximately 20% (Stocker et al., 2013). Here we did not consider the mineral dust effects because the quantification of the dust impact on BC-snow albedo effects is closely depended on the performance in simulating the dust distributions, which is not quite satisfied and associated with large uncertainties in the existing models (Ridley et al., 2016). The mineral dust is mainly emitted from natural sources. The complex mineral dust emission makes it difficult to represent accurately in weather and climate models (Kok, 2011). To avoid the extra uncertainties in our analysis, the dust effects on snow are not explicitly calculated in this study. Similar methods have been implemented in previous studies (He et al., 2014a; Kopacz et al., 2011).

We have clarified the methods we used to calculate the BC-in-snow forcing and added

a discussion on this part of uncertainty in the **Supplementary Information** (P5, L207-213 and P8, L309-312).

“...The underlying ground albedo is obtained from GEOS-5 meteorological fields. The snow albedo is computed via the adding/doubling radiative transfer method using the single scattering properties of the BC-snow mixtures. We fully consider different snow grain shapes (sphere or spheroid), BC-snow mixing states (internal or external), and BC coating states (coated or uncoated) in our estimation, and the mean value is used in the final report. The BC-induced snow albedo reduction is calculated as the difference in the snow albedo both with and without BC.”

[L207-213, Page 5]

“...Additionally, dust deposited on the snow can lower snow albedo, which was not considered in this study to avoid additional uncertainty from dust simulations^{95,96}. The Intergovernmental Panel on Climate Change (IPCC) Fifth Assessment Report indicated that co-existing dust may reduce BC SAF by approximately 20%¹³.”

[L309-312, Page 8]

3. The net forcing is not caused by only black carbon but also organic carbon, sulfate, ozone in addition to CO₂. Is black carbon the dominating forcing? Is it acceptable to disregard the other pollutants? Any change in activity, or trade etc, would affect all of these.

Thanks for your question! In this study, we focus on the BC because of its unique property and important effects in the climate. It is regarded as the dominating forcing for the HTP glacier retreat besides CO₂ (Qian et al., 2015; Ramanathan et al., 2007). The BC-in-snow effects on the glaciers may even exceed that of pre-industrial to present CO₂ increase (Qian et al., 2011). As a unique anthropogenic climate forcer, the property of BC is quite different from other aerosols. The mineral dust and high-altitude ozone are mainly from natural sources that cannot be directly attributed to human activities. Other short-lived aerosols like sulfate and organic carbons exert negative radiative forcing in climate mainly when they are traveling in the atmosphere with a timescale of one week. IPCC AR5 showed that the radiative forcing of sulfate aerosols was estimated to be -0.4 (-0.6 to -0.2) W m^{-2} from (Stocker et al., 2013). The climate effects of organic carbons, including the primary and secondary parts, remain large uncertain in present knowledge. The IPCC AR5 gave an estimate of -0.09 (-0.16 to -0.03) W m^{-2} and -0.03 (-0.27 to $+0.20$) W m^{-2} for the primary organic aerosol and secondary organic aerosol, respectively (Stocker et al., 2013). The effects of sulfate and organic carbon on the HTP glaciers are much less significant than that of BC. Therefore, we decide to focus our discussion on the BC pollutions over the HTP following previous studies (He et al., 2014a; Kopacz et al., 2011; Lu et al., 2012; Zhang et al., 2015).

References:

He, C., Li, Q., Liou, K.-N., Takano, Y., Gu, Y., Qi, L., Mao, Y., Leung, L.R., 2014a.

Black carbon radiative forcing over the Tibetan Plateau. *Geophysical Research Letters* 41, 7806-7813.

He, C., Li, Q.B., Liou, K.N., Zhang, J., Qi, L., Mao, Y., Gao, M., Lu, Z., Streets, D.G., Zhang, Q., Sarin, M.M., Ram, K., 2014b. A global 3-D CTM evaluation of black carbon in the Tibetan Plateau. *Atmospheric Chemistry and Physics* 14, 7091-7112.

Kok, J.F., 2011. A scaling theory for the size distribution of emitted dust aerosols suggests climate models underestimate the size of the global dust cycle. 108, 1016-1021.

Kopacz, M., Mauzerall, D.L., Wang, J., Leibensperger, E.M., Henze, D.K., Singh, K., 2011. Origin and radiative forcing of black carbon transported to the Himalayas and Tibetan Plateau. *Atmospheric Chemistry and Physics* 11, 2837-2852.

Li, C.L., Bosch, C., Kang, S.C., Andersson, A., Chen, P.F., Zhang, Q.G., Cong, Z.Y., Chen, B., Qin, D.H., Gustafsson, O., 2016. Sources of black carbon to the Himalayan-Tibetan Plateau glaciers. *Nature Communications* 7.

Lu, Z., Streets, D.G., Zhang, Q., Wang, S., 2012. A novel back-trajectory analysis of the origin of black carbon transported to the Himalayas and Tibetan Plateau during 1996-2010. *Geophysical Research Letters* 39, n/a-n/a.

Qian, Y., Flanner, M.G., Leung, L.R., Wang, W., 2011. Sensitivity studies on the impacts of Tibetan Plateau snowpack pollution on the Asian hydrological cycle and monsoon climate. *Atmospheric Chemistry and Physics* 11, 1929-1948.

Qian, Y., Yasunari, T.J., Doherty, S.J., Flanner, M.G., Lau, W.K.M., Ming, J., Wang, H., Wang, M., Warren, S.G., Zhang, R.J.A.i.A.S., 2015. Light-absorbing particles in snow and ice: Measurement and modeling of climatic and hydrological impact. 32, 64-91.

Ramanathan, V., Chung, C., Kim, D., Bettge, T., Buja, L., Kiehl, J.T., Washington, W.M., Fu, Q., Sikka, D.R., Wild, M., 2005. Atmospheric brown clouds: Impacts on South Asian climate and hydrological cycle. *Proceedings of the National Academy of Sciences of the United States of America* 102, 5326-5333.

Ramanathan, V., Ramana, M.V., Roberts, G., Kim, D., Corrigan, C., Chung, C., Winker, D., 2007. Warming trends in Asia amplified by brown cloud solar absorption. *Nature* 448, 575-U575.

Ridley, D.A., Heald, C.L., Kok, J.F., Zhao, C., 2016. An observationally constrained estimate of global dust aerosol optical depth. *Atmospheric Chemistry and Physics* 16, 15097-15117.

Stocker, T., Qin, D., Plattner, G., Tignor, M., Allen, S., Boschung, J., Nauels, A., Xia, Y., Bex, B., Midgley, B., 2013. IPCC, 2013: climate change 2013: the physical science basis. Contribution of working group I to the fifth assessment report of the intergovernmental panel on climate change. Cambridge University Press.

Zhang, R., Wang, H., Qian, Y., Rasch, P.J., Easter, R.C., Ma, P.L., Singh, B., Huang, J., Fu, Q., 2015. Quantifying sources, transport, deposition, and radiative forcing of black carbon over the Himalayas and Tibetan Plateau. *Atmospheric Chemistry and Physics* 15, 6205-6223.

Response to Anonymous Reviewer 4:

(Note: Reviewer comments are listed in grey, and responses to reviewer comments are in black. Pasted text from the new version of the paper is in italics.)

The title of this manuscript was interesting that linked the glaciers retreat with global trade. Many recent studies indicated the black carbon (or other pollutions) originated from South Asia and the rest regions outside of the Third Pole (TP) could transport to the plateau and impacted on the local climate, using different approaches including monitoring network, geochemical evidence and combination of modeling technology. More results showed India should take responsibility for glaciers melting in the TP induced by anthropogenic BC, however, ignored the potential transfer BC emission from USA and Europe through global trade. I generally agreed the authors' viewpoints and thought the results were novel. Furthermore, the manuscript needs substantial modification until reaching to the criteria.

We really appreciate the reviewer's thoughtful and positive comments. We believe the manuscript has improved a lot by addressing the concerns below. Please see our response to each comment below.

In the title referred to the "The third Pole", this domain was a great region not only involving the Tibetan Plateau and Himalayas, but also including Tianshan Mt., Qilian Mt., Kunlun Mt. and the others mountain regions over the Xinjiang, Yunnan, and Sichuan in China. Actually, HTP \neq Third Pole. So the authors should consider more glacier areas as showed in Fig. 1 blue rectangles.

Thanks for your suggestion. We agree that the Third Pole includes more regions besides the Himalayas and Tibetan Plateau (HTP). However, the distribution of the Third Pole glaciers is quite complex. The glacier areas and their relative importance are significantly different for various grids within the domain. Simply considering all regions over the Third Pole may receive substantial criticism towards its policy implications. Therefore, we used a determinative condition to select the receptor boxes for adjoint simulations. The blue rectangles in Fig.1 represent the model grids where the glacier area in each grid exceeds 2 % based on the World Glacier Inventory (WGI). We understand the reviewer's concern about the mountains that were not included in our analysis. Since our focus is mainly on the Himalayas and Tibetan Plateau, which is the major domain of the Third Pole and plays a critical role in the Asian hydrological cycle and climate system, we have changed the title of this manuscript to "***How Do Global Trade Cascade to Large Climate Forcing over the Tibetan Plateau Glaciers***" in the revised version.

Figure 1 | The spatial distribution of glacier areas over the HTP and adjacent regions. The rectangles denote the selected HTP glacier regions defined as receptors in the adjoint simulation, where the glacier area in each grid exceeds 2 % based on the World Glacier Inventory (WGI) (WGMS and NSIDC, 1989, updated 2012). Black pentagrams are 17 measurement sites of the surface BC concentrations: Nainital (b), Kharagpur (b), Kanpur (c), Gandhi College (d), Nagarkot (e), Langtang (f), Nepal Climate Observatory at Pyramid (NCOP) (g), Manora Peak (h), Nam Co Observational Station (NCOS) (i), Zhuzhang (j), Muztagh Ata (k), Hanle (l), Lutang (m), QOMS (n), Beiluhe (o), Ranwu (p) and QSSGEE (q). Black circles are 20 measurement sites of BC concentrations in the snow: Zuoqiupu (A), Qiangyong (B), Noijin Kangsang (C), East Rongbuk (D), Kangwure (E), Namunani (F), Mt. Muztagh (G), Laohugou #12 (H), Qiyi (I), 1 July glacier (J), Meikuang (K), Dongkemadi (L), La'nong (M), Zhadang (N), Haxilegen River (O), Urumqi Riverhead (P), Miao'ergou #3 (Q), Demula glacier (R), Muji glacier (S) and Tianshan Urumpi glacier #1 (T). Please see the Supplementary Information for additional details.

The method was not clearly written. Specifically, how did you obtain the seasonal mean BC concentration contributed by consumption-based from India and China, as showed in Fig. s5. I suggested to provide a framework in SI to help reader understanding the steps.

Thanks for your constructive suggestion! A framework for the methods used in this study has been added in the **Supplementary Information** (i.e., Supplementary Figure 1). The following sentences in the main text have also been revised to describe the steps of our analysis (P3, L85-98). We have also reorganized the **Materials and Methods** section to provide more details about our analysis (please refer to the revised manuscript). In this study, we firstly use the adjoint model to calculate the $2^{\circ} \times 2.5^{\circ}$ source-receptor sensitivity map of BC over the HTP. By multiplying the sensitivity with the global emissions, we can estimate the transport efficiency of the BC emissions from each grid cell to the HTP glaciers. Then, we use an MRIO model to estimate global BC emissions embodied by the global trade, i.e., the consumption-based emissions. The MRIO model determines where goods and services are

ultimately consumed domestically based on detailed data of the economic structure describing the product exchanges within and among regions and hence can quantitatively assign emissions produced in one region upstream along the supply chain to various regions as final consumers. The production- and consumption-based emissions in different regions are further combined with the emission sensitivity derived from adjoint simulations to estimate the relative contributions of different regions to the BC over the HTP from production and consumption perspectives. The radiative forcing (i.e., DRF and SAF) of BC over the HTP are calculated using a radiative transfer model and a stochastic snow albedo model. These radiative forcing are further assigned to different countries/sectors according to their relative contributions to the BC pollutions over the HTP from production and consumption perspectives.

Additionally, the seasonal mean BC concentrations contributed by consumption-based emissions from India and China are obtained from model simulations. We conducted two sensitivity simulations using the GEOS-Chem forward model in which the consumption-based BC emissions from China and India are removed, following previous studies like Lin et al. (2016). Fig.S7 (Fig.S5 in the original version) shows the corresponding changes in BC surface concentrations over the HTP and adjacent regions. These sensitivity simulations help us compare the performance of the adjoint analysis with the forward model approach. We have added the following sentence (P8, L287-L290) in the method to make it clear. The relevant result has been discussed in the **Supplementary Information** (P7, L269-278)

In main text:

“To address this issue, we use the adjoint of the Goddard Earth Observing System (GEOS)-Chem model to identify the locations from which the BC currently situated over the glaciers of the HTP originate and to quantify their relevant emission sensitivities. We further trace the flow of direct geographic BC emissions along the supply chain forward to final consumers using a multi-region input-output (MRIO) model. The production- and consumption-based emissions are then combined with the emission sensitivities derived from adjoint simulations to estimate the relative contributions of different countries/regions to BC pollutions over the HTP from production and consumption perspectives, respectively. The corresponding direct radiative forcing (DRF) and snow albedo forcing (SAF) of BC over the HTP glaciers are calculated using a radiative transfer model and a stochastic snow albedo model. The objective of this study is to provide a combined assessment of the sources and radiative forcing of BC over the HTP glaciers from multiple perspectives (Supplementary Figure 1).”

[L85-98 Page 3]

“...To evaluate the performance of adjoint simulations, we further conduct two sensitivity simulations using the GEOS-Chem forward model in which the consumption-

based BC emissions from China and India are removed, respectively (Supplementary Figure 7).”

[L287-290, Page 8]

In the **Supplementary Information**:

“... We conduct two sensitivity simulations using the GEOS-Chem forward model in which the consumption-based BC emissions from China and India are removed. Fig.S7 shows the corresponding changes in the BC surface concentrations over the HTP and adjacent regions. The results reveal that removing the consumption-based BC emissions of China and India will reduce the BC column concentrations over the HTP glaciers by 19.2 % and 28.9 %, respectively. These results are very close to our estimations using the combined adjoint and MRIO analysis, in which the relative contributions from China and India from the consumption perspective are 18.4 % and 30.7 %, respectively.”

[L269-278, Page 7]

Supplementary Figure 1 | Schematic methodology for the combined adjoint and multi-region input-output analysis

Supplementary Figure 7 | The seasonal-mean BC surface concentrations over the HTP and adjacent regions contributed by consumption-based emissions from India (a-d) and China (e-h). These results are estimated using the GEOS-Chem forward model simulations. We use the GEOS-Chem forward model to conduct a base simulation in which all BC emissions are included and two sensitivity simulations in which the consumption-based BC emissions from China and India are removed. Changes in BC surface concentrations of these sensitivity simulations comparing to the base simulation are showed here.

GEOS-CHEM was used to calculate the contribution from the source region, but its resolution was very coarse with $2 \times 2.5^\circ$ (may be more). As showed in Fig.2, I saw too much mass burden over the Southern TP and Himalayas, however, actually there was no or few emissions derived from the local activities. I trust such “biases” was induced by the coarse resolution. The authors mentioned a process that transfer the grid of $2 \times 2.5^\circ$ to $0.1 \times 0.1^\circ$, please provide the details. Did you consider any other factor such as terrain effect in the transfer process? Otherwise the interpolation of such downscale did not make sense.

Thanks for your question! In the study, we use the GEOS-Chem model to simulate the BC transport from different sources to the HTP. The BC simulation in GEOS-Chem is widely used in previous studies and its performance has been evaluated extensively in these studies (He et al., 2014b; Kopacz et al., 2011). Among them, He et al. (2014b) has provided a systematic evaluation of BC simulations in GEOS-Chem over the HTP. In this study, we further evaluated its performance using more updated in situ measurements of BC in the surface air and snow over the HTP and adjacent regions. These results all demonstrated that the GEOS-Chem model exhibits a satisfying performance in the simulation of the atmospheric transport and deposition of BC over the HTP. It provides us the confidence to use this model framework in our analysis. Previous studies interested in the source attributions of BC over the HTP have used similar resolutions in their model simulations (Kopacz et al., 2011; Lu et al., 2012; Zhang et al., 2015). We agree with the reviewer that the GEOS-Chem model has limitations in describing the distributions of BC emitted within the grid as well as the meteorological fields for the adjacent regions beneath the HTP due to the complex terrain effects, which is unfortunately an inherent problem existed in most of chemistry transport models. Hence, it may overestimate the contributions from the adjacent regions beneath the Himalayas to the HTP.

In this study, we downscaled the $2^\circ \times 2.5^\circ$ source-receptor sensitivity map of BC over the HTP to a smaller resolution map based on a high-resolution emission data set. The PKU-BC-Inventory provides us a $0.1^\circ \times 0.1^\circ$ emission inventory of BC. The contribution of a $2^\circ \times 2.5^\circ$ model grid derived from the adjoint analysis was further attributed to a set of $0.1^\circ \times 0.1^\circ$ grid according to their emission intensities. Similar approaches have been used in previous studies (Xu et al., 2018; Zhang et al., 2009b). It is based on a basic principle that regions emitted more BC may have more contributions to the HTP and should share more responsibilities. Although this approach does not directly consider the terrain effects, which has not been well understood yet, it considers the sub-grid distribution of BC emissions and covers the shortage of the model resolution more or less. Figure 2 shows the distributions of original results from the adjoint analysis. It is notable that the overestimate over the Southern TP and Himalayas has been significantly improved after the downscale approach.

We have provided more details about the downscale approach in the revised manuscript (see P8, L274-280). The relevant uncertainties have been discussed in the

Supplementary Information (see P7, L284-292).

In the manuscript:

“...The $2^\circ \times 2.5^\circ$ source-receptor sensitivity map of BC in the HTP derived from the adjoint analysis is disaggregated to $0.1^\circ \times 0.1^\circ$ using high-resolution emission inventory as a proxy. The emission intensity at a $0.1^\circ \times 0.1^\circ$ resolution is obtained from the PKU-BC-Inventory (Wang et al., 2014) (Supplementary Figure 2). This approach is based on a basic principle that regions emitted more BC may have more contributions to the HTP and should share more responsibilities. It considers the sub-grid distribution of BC emissions and hence covers the shortage of the model resolution more or less.”

In the **Supplementary Information**:

“...In this study, we downscaled the $2^\circ \times 2.5^\circ$ source-receptor sensitivity map of BC over the HTP to a finer resolution using emission as a proxy, which assumes that the transport efficiency is the same within a model grid. Considering that the long-distance atmospheric transport of BC to the HTP glaciers are following the large-scale circulations in the atmosphere, the approach has negligible effects on the estimation of most countries/regions. As for the adjacent regions like Nepal, which are beneath the HTP, the sub-grid transport efficiency may be different due to the complex terrain effects. Hence, it may contribute to the uncertainties in estimations of these regions.”

Another question was the connection between the modeling approach of MRIO and GEOS-CHEM, please clarify. The authors suggested that the virtual transport pathway was more efficient than the traditional atmospheric pathway. Does that mean the traditional pathway might be underestimated? If it was true, how was the updated contributed rate (the traditional atmospheric pathway were approximately 68.9% and 16.3% from South Asia and China respectively) once the virtual pathway was considered.

Thanks for your questions! The chemistry transport models like GEOS-Chem simulate the transport of BC directly emitted from surface sources (i.e., the production-based emissions) to other regions following atmospheric transport pathways. The efficiencies of these transport pathways depend on the wind strength as well as the residential time of BC in the atmosphere. On the other hand, the geographical separation of production and consumption following global trade leads to a shift in air pollutant emissions and their associated environmental pressures among various regions. Part of BC directly emitted from one country/region can be induced by the consumptions of other countries/regions through the import of goods and services. The MRIO model can estimate the BC emissions embodied in global trade (i.e., the consumption-based emissions) and hence identifies a virtual transport pathway of BC among different

regions. This virtual transport pathway is embodied in the interregional exchange of commodities. Different from the atmospheric transport pathways, this virtual transport pathway represents interregional flows of air pollutants embodied within global trade that may transfer from one region to the others through economic activities. The efficiency of the virtual transport pathway closely depends on the scales of the commodities import/export within one country/region. Since this efficiency is directly related to the emission transfer, it is much larger than that of traditional atmospheric transport pathways (Lin et al., 2014; Meng et al., 2018; Zhang et al., 2017a)

In our analysis, we find that global trade transfer BC emissions from remote regions like US and Europe to Asian regions where the HTP is more vulnerable to. Regions like USA and Europe may exert nearly ten times larger effects on the HTP in global trade than their direct contributions through atmospheric transport pathways. As for the South Asia and China, their contributions to the HTP become smaller when we consider the effects of global trade. The relative contribution of China (outside the HTP) to BC over the HTP decreases from 16.3% to 14.6% correspondently. The relative contribution of India also decreases from 30.1% to 28.5%. The changes in the relative contributions of different regions to the BC over the HTP glaciers when we consider the virtual transport pathway are shown in Supplementary Figure 10. In the revised manuscript, we have provided more details about this issue (P4-5, L140-164).

“...Figure 3a shows the virtual interregional BC flows embodied in the import/export of goods and services among various countries/regions (see Supplementary Figure 9 for a map of the defined regions). China, which is a primary BC source region, accounts for 30 % of the global anthropogenic emissions. More than 10 % of its direct emissions (above 0.2 Tg) can be attributed to products and services consumed by other countries/regions, mainly the USA and Europe (Figure 3a). Considerable BC emissions in India and Southeast Asia are also related to final consumption in other regions. The consumption of Middle Asia enables the downstream transport of BC emissions (0.06 Tg) to primarily China and Europe. Africa and Southeast Asia export their products and services to remote regions like Europe while consume products and services mainly from China. Even though the effects of direct BC emissions from the USA and Europe on the HTP glaciers are negligible, the international trade transfers large volumes of BC emissions from the USA and Europe to China, India and other Asian regions, and those emissions may be subsequently transported to the HTP (Figure 3a).

The relative contributions of different regions to the BC over the HTP glaciers from production- and consumption-based perspectives are shown in Supplementary Figure 10. The USA and Europe play significantly more important roles in the BC pollution

over the HTP glaciers as final consumers than as direct emitters. Our estimation shows that nearly 10 % of the BC over the HTP glaciers are embodied by interregional flow within global trade though the importation of commodities. Generally, the global trade leads to a virtual transport pathway of BC from remote regions to the HTP glaciers. This process aggravates the BC pollution towards the HTP glaciers. ”

Supplementary Figure 10 | Relative contributions (%) from different countries/regions to the BC over the HTP glaciers from production- and consumption--based perspectives. a, Relative contributions (%) from different countries/regions to the BC over the HTP glaciers according to the production-based accounting method. **b,** Differences in the results from the consumption-based perspectives relative to those from the production-based perspective. Please refer to Data S3 for details.

The BC inventory was a probably uncertainty factor both in GEOS-CHEM and MRIO. I found only a dataset called “PKU-BC-inventory” was input. Even though the data was well agreement with observation, it was preferable to compare with another dataset of inventory applied in the model. In this way, the results trend to be more solid.

Thanks for your suggestion! In this study, we conducted several experiments using different emission inventories including the default anthropogenic BC emission inventory from Bond et al. (2007), an updated anthropogenic BC emission inventories from Lu et al. (2011) for China and India and those from Zhang et al. (2009a) for the rest of Asia, as well as a global BC emission inventory from PKU-BC-Inventory

(Wang et al., 2014). The Supplementary Table 1 shows the details of our model experiments. The PKU-BC-Inventory showed a better performance in simulating BC concentrations in the surface air and snow in the model comparing with other inventories. The PKU-BC-Inventory is based on an updated fuel consumption and emission factor dataset. A detailed comparison between this inventory and previous inventories has been provided in Wang et al. (2014). What's more, the PKU-BC-Inventory provides us a detailed BC emission data set covering 64 fuel combustion processes and 14 industrial processes throughout 233 countries/territories, which has great advantage in conducting the MRIO analysis. Therefore, we decided to use the PKU-BC-Inventory in the MRIO analysis.

Supplementary Table 1 |List of GEOS-Chem model experiments

Experiment	A	B	C	D	E
			China and India: Lu et al. (2011)		
Anthropogenic emissions	Bond et al. (2007)	Bond et al. (2007)	Rest of Asia: Zhang et al. (2009) Rest of world: Bond et al. (2007)	Wang et al. (2014)	Wang et al. (2014)
Biomass burning emissions			GFEDv3 (van der Werf et al., 2010)		
BC ageing	e-folding time 1.15 days	He et al. (2016)	e-folding time 1.15 days	e-folding time 1.15 days	He et al. (2016)

For model evaluation, the current collections from previous studies were not the updated data set (most data were for 2004-2006 in Supplementary Table 3, S6). In fact, recent years many publications reported the observations in terms of BC concentrations both in the atmosphere and in snow as well as its albedo effect over the Third Pole.

The most updated data and publications must be considered, such as:

Li X, et al. (Light-absorbing impurities accelerate glacier melt in the Central Tibetan Plateau. *Science of the Total Environment*, 2017, 587: 482-490.) reported the data in the Central of Tibetan Plateau.

Niu, H., et al. (In-situ measurements of light-absorbing impurities in snow of glacier on Mt. Yulong and implications for radiative forcing estimates. *Science of The Total Environment*, 2017, 581, 848-856.) reported the data in the southeastern Tibetan Plateau.

Zhang, Y., et al. (Light-absorbing impurities enhance glacier albedo reduction in the southeastern Tibetan plateau. *Journal of Geophysical Research: Atmospheres*, 2017, 122. Doi: 10.1002/2016JD026397.) reported the data in the southern Tibetan Plateau.

Zhang, Y., et al. (Characteristics of black carbon in snow from Laohugou No. 12 glacier on the northern Tibetan Plateau. *Science of the Total Environment*, 607, 1237-1249.) reported the data in the northeastern Tibetan Plateau.

Zhang, Y., et al. (Black carbon and mineral dust in snow cover on the Tibetan Plateau. *Cryosphere*, 2018, 12(2), 413-431.) reported more data of BC-in-snow cover the region of Tibetan Plateau.

Zhang Y. L. et al. (Research progress of light-absorbing impurities in glaciers of the Tibetan Plateau and its surroundings (in Chinese). *Chin. Sci. Bull.*, 2017, 62(35): 4151-4162. Doi: 10.1360/N972017-00505)) provided a review BC and other LAIs in the Third Pole.

Chen X. et al. (Concentration, temporal variation, and sources of black carbon in the Mt. Everest region retrieved by real-time observation and simulation. *Atmos. Chem. Phys.*, 2018, 18, 12859-12875. Doi: 10.5194/acp-18-12859-2018) reported two year BC in the atmosphere on Everest Station.

Good suggestion! We have added more updated observations of BC in the surface air and snow to evaluate our model results (Chen et al., 2018; Cong et al., 2015; Li et al., 2017; Ming et al., 2016; Qu et al., 2014; Wan et al., 2015; Wang et al., 2016; Yang et al., 2015; Zhang et al., 2017b; Zhao et al., 2012) (shown in Supplementary Table 2 and S3). Most of observations suggested by the reviewer have been included here except for observations of BC in the aged snow, considering that observed BC concentrations in the aged snow may vary by 1-3 orders of magnitude (Niu et al., 2017; Zhang et al., 2017c). Comparisons with available observations demonstrate that the model exhibits a good performance in simulating the atmospheric transport and deposition of BC over the HTP. Please refer to the revised manuscript for more details.

Supplementary Table 2 | List of measurement sites of surface BC concentrations used in this study for the model evaluation

Site	Lat(°N)	Long (°E)	Elev.(m)	Time	Technique	Source
Nainital	29.2	79.3	1950	2006 Jan-May	Aethalometer	Beegum et al.(2009)
Kharagpur	22.5	87.5	28	2005-2007	Aethalometer	Nair et al.(2012)
Kanpur	26.5	80.3	142	2007-2008	TOT	Ram et al.(2010b)
Gandhi College	25.9	84.1	158	Sep.-Dec. 2006	retrieval	Ganguly et al.(2009b)
Nagarkot	27.7	85.5	2150	1999-2000	TOT	Carrico et al.(2003)
Langtang	28.1	85.6	3920	1999-2000	TOT	
NCOP	28.0	86.8	5079	2006-2008	MAAP	Bonasoni et al.(2010)
Manora Peak	29.4	79.5	1950	2005-2008	TOT	Ram et al.(2010a)
NCOS	30.8	91.0	4730	2006	TOR	Ming et al.(2010)
				2012	TOR	Wan et al.(2015)
Zhuzhang	28.0	99.7	3583	Aug.2004-Feb.2005	TOR	Qu et al.(2008)

Muztagh Ata	38.3	75.0	4500	2003-2006	TOR	Cao et al. (2009)
Hanle	32.8	79.0	4250	2009-2010	Aethalometer	Babu et al. (2011)
Lulang	29.5	94.4	3300	2008-2009	TOR	Zhao et al. (2013)
QOMS	28.4	87.0	4276	Aug. 2009-Jul. 2010	TOR	Cong et al.(2015)
				May 2015-May 2017	Aethalometer	Chen et al.(2018)
Beiluhe	34.9	92.9	4600	Nov. 2012-Jun. 2013	Aethalometer	Wang et al.(2016)
Ranwu	29.3	97.0	4600	Nov. 2012-Jun. 2013	Aethalometer	
QSSGEE	39.5	96.5	4214	May 2009-Mar. 2011	Aethalometer	Zhao et al.(2012)

*TOT: thermal-optical transmittance; MAAP: multi-angle absorption photometer; TOR thermal-optical reflectance.

Supplementary Table 3 | List of measurement sites of the BC concentration in snow used in this study for the model evaluation

Site	Lat(°N)	Long (°E)	Elev.(km)	Time	Source
Zuoqiupu	29.21	96.92	5.5	monsoon 2006	Xu et al.(2009)
	29.21	96.92	5.6	non-monsoon 2006	
Qiangyong	28.83	90.25	5.4	summer 2001	Xu et al.(2006)
Noijin Kangsang	29.04	90.2	5.95	annual 2005	Xu et al.(2009)
East Rongbuk	28.02	86.96	6.5	monsoon 2001	Ming et al.(2008)
	28.02	86.96	6.5	non-monsoon 2001	
	28.02	86.96	6.5	summer 2002	Ming et al.(2009a)
	28.02	86.96	6.5	Oct.2004	
	28.02	86.96	6.5	Sept.2006	
	28.02	86.96	6.52	May 2007	
Kangwure	28.47	85.82	6	summer 2001	
Namunani	30.45	81.27	5.9	summer 2004	Xu et al.(2006)
Mt.Muztagh	38.28	75.02	6.35	summer 2001	
	38.28	75.1	6.3	1999	
Laohugou #12	39.43	96.56	5.05	Oct.2005	Ming et al.(2009a)
Qiyi	39.23	97.06	4.85	Jul.2005	Ming et al.(2010)
1 July glacier	39.23	97.75	4.6	summer 2001	Xu et al.(2006)
Meikuang	35.67	94.18	5.2	Nov.2005	Ming et al.(2009b)
Dongkemadi	33.1	92.08		summer 2001	Xu et al.(2006)
	33.1	92.08	5.6	year 2005	Ming et al.(2013)
	33.1	92.08		Aug.2014-Oct.2015	Li et al.(2017)
La'nong	30.42	90.57	5.85	Jun.2005	Ming et al.(2009a)
Zhadang	30.47	90.5	5.5-5.8	Jul.2006	Ming et al.(2010)
	30.47	90.5	5.8	Jul.-Aug. 2012	Qu et al.(2014)
Haxilegen River	43.73	84.46	3.76	Oct.2006	Ming et al.(2011a)
Urumqi Riverhead	43.1	86.82	4.05	Nov.2006	Ming et al.(2009b)

Miao'ergou #3	43.06	94.32	4.51	Aug.2005	Ming et al.(2009a)
Demula glacier	29.35	97.2	5.1	Jun.2015	Zhang et al.(2017)
Muji glacier	39.19	73.74	5.5	Jul.2012	Yang et al.(2015)
Tianshan Urumpi glacier #1	43.1	86.82	3.9	Aug. 2013	Ming et al.(2016)

Minor suggestions for references.

In Line 71, citations 2, 6 indicated the increased BC could induce rapid glacier melting, but did not referred or showed the evidence the BC being transported into the HTP. One study based on the ¹⁴C isotope technology (Li C. et al., Sources of Black Carbon to the Himalayan-Tibetan Plateau glaciers. Nature Communications, 2016, 7: 12574. Doi: 10.1038/ncomms12574.) has provided the geochemical evidence.

We have added this reference to support out statement (P2, L63-L65).

“...Ample evidence shows that the increasing amount of BC being transported into the HTP plays an important role on the observed rapid glacial retreat therein comparable to the effects of greenhouse gases (GHGs) (Li et al., 2016; Menon et al., 2010; Xu et al., 2009).”

Line77, “Various approaches...” were not various, actually all were based on modeling technology.

We have rephrased the sentence (P3, L73):

“Previous studies have made efforts to identify the origins of BC over the HTP during the last decade (He et al., 2014a; Kopacz et al., 2011; Lu et al., 2012; Zhang et al., 2015).”

References:

- Bond, T.C., Bhardwaj, E., Dong, R., Jogani, R., Jung, S., Roden, C., Streets, D.G., Trautmann, N.M., 2007. Historical emissions of black and organic carbon aerosol from energy-related combustion, 1850–2000. *Global Biogeochemical Cycles* 21.
- Chen, X.T., Kang, S.C., Cong, Z.Y., Yang, J.H., Ma, Y.M., 2018. Concentration, temporal variation, and sources of black carbon in the Mt. Everest region retrieved by real-time observation and simulation. *Atmospheric Chemistry and Physics* 18, 12859-12875.
- Cong, Z., Kang, S., Kawamura, K., Liu, B., Wan, X., Wang, Z., Gao, S., Fu, P., 2015. Carbonaceous aerosols on the south edge of the Tibetan Plateau: concentrations, seasonality and sources. *Atmos. Chem. Phys.* 15, 1573-1584.
- He, C., Li, Q., Liou, K.N., Takano, Y., Gu, Y., Qi, L., Mao, Y., Leung, L.R., 2014a. Black carbon radiative forcing over the Tibetan Plateau. *Geophysical Research Letters* 41, 7806-7813.
- He, C., Li, Q.B., Liou, K.N., Zhang, J., Qi, L., Mao, Y., Gao, M., Lu, Z., Streets, D.G., Zhang, Q., Sarin, M.M., Ram, K., 2014b. A global 3-D CTM evaluation of black carbon in the Tibetan Plateau. *Atmospheric Chemistry and Physics* 14, 7091-7112.

- Kopacz, M., Mauzerall, D.L., Wang, J., Leibensperger, E.M., Henze, D.K., Singh, K., 2011. Origin and radiative forcing of black carbon transported to the Himalayas and Tibetan Plateau. *Atmospheric Chemistry and Physics* 11, 2837-2852.
- Li, C.L., Bosch, C., Kang, S.C., Andersson, A., Chen, P.F., Zhang, Q.G., Cong, Z.Y., Chen, B., Qin, D.H., Gustafsson, O., 2016. Sources of black carbon to the Himalayan-Tibetan Plateau glaciers. *Nature Communications* 7.
- Li, X.F., Kang, S.C., He, X.B., Qu, B., Tripathee, L., Jing, Z.F., Paudyal, R., Li, Y., Zhang, Y.L., Yan, F.P., Li, G., Li, C.L., 2017. Light-absorbing impurities accelerate glacier melt in the Central Tibetan Plateau. *Sci Total Environ* 587, 482-490.
- Lin, J., Pan, D., Davis, S.J., Zhang, Q., He, K., Wang, C., Streets, D.G., Wuebbles, D.J., Guan, D., 2014. China's international trade and air pollution in the United States. *Proceedings of the National Academy of Sciences of the United States of America* 111, 1736.
- Lin, J., Tong, D., Davis, S., Ni, R., Tan, X., Pan, D., Zhao, H., Lu, Z., Streets, D., Feng, T., 2016. Global climate forcing of aerosols embodied in international trade. *Nature Geoscience* 9, 790-794.
- Lu, Z., Streets, D.G., Zhang, Q., Wang, S., 2012. A novel back-trajectory analysis of the origin of black carbon transported to the Himalayas and Tibetan Plateau during 1996-2010. *Geophysical Research Letters* 39, n/a-n/a.
- Lu, Z., Zhang, Q., Streets, D.G., 2011. Sulfur dioxide and primary carbonaceous aerosol emissions in China and India, 1996–2010. *Atmospheric Chemistry and Physics* 11, 9839-9864.
- Meng, J., Liu, J., Yi, K., Yang, H., Guan, D., Liu, Z., Zhang, J., Ou, J., Dorling, S., Mi, Z., Shen, H., Zhong, Q., Tao, S., 2018. Origin and Radiative Forcing of Black Carbon Aerosol: Production and Consumption Perspectives. *Environmental Science & Technology* 52, 6380-6389.
- Menon, S., Koch, D., Beig, G., Sahu, S., Fasullo, J., Orlikowski, D., 2010. Black carbon aerosols and the third polar ice cap. *Atmospheric Chemistry and Physics* 10, 4559-4571.
- Ming, J., Xiao, C., Wang, F., Li, Z., Li, Y.J.E.S., Research, P., 2016. Grey Tianshan Urumqi Glacier No.1 and light-absorbing impurities. 23, 9549-9558.
- Niu, H.W., Kang, S.C., Shi, X.F., Paudyal, R., He, Y.Q., Li, G., Wang, S.J., Pu, T., Shi, X.Y., 2017. In-situ measurements of light-absorbing impurities in snow of glacier on Mt. Yulong and implications for radiative forcing estimates. *Sci Total Environ* 581, 848-856.
- Qu, B., Ming, J., Kang, S.C., Zhang, G.S., Li, Y.W., Li, C.D., Zhao, S.Y., Ji, Z.M., Cao, J.J., 2014. The decreasing albedo of the Zhadang glacier on western Nyainqentanglha and the role of light-absorbing impurities. *Atmos. Chem. Phys.* 14, 11117-11128.
- Wan, X., Kang, S., Wang, Y., Xin, J., Liu, B., Guo, Y., Wen, T., Zhang, G., Cong, Z., 2015. Size distribution of carbonaceous aerosols at a high-altitude site on the central Tibetan Plateau (Nam Co Station, 4730ma.s.l.). *Atmospheric Research* 153, 155-164.
- Wang, M., Xu, B., Wang, N., Cao, J., Tie, X., Wang, H., Zhu, C., Yang, W., 2016. Two distinct patterns of seasonal variation of airborne black carbon over Tibetan Plateau. *Sci Total Environ* 573, 1041-1052.
- Wang, R., Tao, S., Balkanski, Y., Ciais, P., Boucher, O., Liu, J., Piao, S., Shen, H., Vuolo, M.R., Valari, M., 2014. Exposure to ambient black carbon derived from a unique inventory and high-resolution model. *Proceedings of the National Academy of Sciences* 111, 2459-2463.
- WGMS, NSIDC, 1989, updated 2012. World Glacier Inventory. Compiled and made available by the World Glacier Monitoring Service, Zurich, Switzerland, and the National Snow and Ice Data Center, Boulder CO, U.S.A.
- Xu, B., Cao, J., Hansen, J., Yao, T., Joswita, D.R., Wang, N., Wu, G., Wang, M., Zhao, H., Yang, W., 2009.

Black soot and the survival of Tibetan glaciers. *Proceedings of the National Academy of Sciences* 106, 22114-22118.

Xu, Y., Shen, H., Yun, X., Gao, F., Chen, Y., Li, B., Liu, J., Ma, J., Wang, X., Liu, X., Tian, C., Xing, B., Tao, S., 2018. Health effects of banning beehive coke ovens and implementation of the ban in China. *115*, 2693-2698.

Yang, S., Xu, B., Cao, J., Zender, C.S., Wang, M., 2015. Climate effect of black carbon aerosol in a Tibetan Plateau glacier. *Atmospheric Environment* 111, 71-78.

Zhang, Q., Jiang, X., Tong, D., Davis, S.J., Zhao, H., Geng, G., Feng, T., Zheng, B., Lu, Z., Streets, D.G., 2017a. Transboundary health impacts of transported global air pollution and international trade. *Nature* 543, 705-709.

Zhang, Q., Streets, D.G., Carmichael, G.R., He, K., Huo, H., Kannari, A., Klimont, Z., Park, I., Reddy, S., Fu, J., 2009a. Asian emissions in 2006 for the NASA INTEX-B mission. *Atmospheric Chemistry and Physics* 9, 5131-5153.

Zhang, R., Wang, H., Qian, Y., Rasch, P.J., Easter, R.C., Ma, P.L., Singh, B., Huang, J., Fu, Q., 2015. Quantifying sources, transport, deposition, and radiative forcing of black carbon over the Himalayas and Tibetan Plateau. *Atmospheric Chemistry and Physics* 15, 6205-6223.

Zhang, Y., Tao, S., Shen, H., Ma, J., 2009b. Inhalation exposure to ambient polycyclic aromatic hydrocarbons and lung cancer risk of Chinese population. *106*, 21063-21067.

Zhang, Y.L., Kang, S.C., Cong, Z.Y., Schmale, J., Sprenger, M., Li, C.L., Yang, W., Gao, T.G., Sillanpaa, M., Li, X.F., Liu, Y.J., Chen, P.F., Zhang, X.L., 2017b. Light-absorbing impurities enhance glacier albedo reduction in the southeastern Tibetan plateau. *J Geophys Res-Atmos* 122, 6915-6933.

Zhang, Y.L., Kang, S.C., Li, C.L., Gao, T.G., Cong, Z.Y., Sprenger, M., Liu, Y.J., Li, X.F., Guo, J.M., Sillanpaa, M., Wang, K., Chen, J.Z., Li, Y., Sun, S.W., 2017c. Characteristics of black carbon in snow from Laohugou No. 12 glacier on the northern Tibetan Plateau. *Sci Total Environ* 607, 1237-1249.

Zhao, S., Ming, J., Xiao, C., Sun, W., Qin, X., 2012. A preliminary study on measurements of black carbon in the atmosphere of northwest Qilian Shan. *Journal of Environmental Sciences* 24, 152-159.

Reviewers' comments:

Reviewer #2 (Remarks to the Author):

Suggested outcome: publication in another journal

The article has undergone substantial improvement since the last revision, and I would like to thank the authors for their work and the thoughtful consideration of all of the reviewers' comments.

My comments were all addressed, and in my opinion the article reads much better. I do uphold my concerns about clear denominations of the different quantities that I already mentioned in my previous review. I have made a few specific detailed comments where these inconsistencies are apparent. I believe that the article still lacks a proficient use of the right denominations for each quantity that is discussed.

While I think that at this point the article is on track to be ready for publication, I do suggest that the authors publish the article in a different journal. The basic take-away message from the Discussion section is that those countries that are located close to the HTP (and that would suffer the most from the glacial retreat) bear the main responsibility for the black carbon emissions above the HTP.

This is a result that has been known before (and the authors also discuss this in the beginning). The recommendations are focused on improving combustion efficiency in the neighbouring countries of the HTP, which is an outcome that does not require extensive trade-link analyses.

The discussion also features that the trade-related impacts on BC over the HTP are higher than the direct pathways of emissions originating within the trade partners' border. But this is more of a qualitative statement, and it is also not particularly surprising given the short lifespan to BC in the atmosphere. A quantitative assessment of direct impacts/trade related impacts and the implication of these in the recommendations is not included in the Discussion section.

Therefore, I cannot see the strong research contribution or outcome in this paper that would in my view justify a publication in Nature Communications.

Detailed review:

Line 124:

I assume that "relative contribution from China" relates to the production-based emissions. I would suggest you use consistent naming of the quantities.

Line 147:

You are using the phrase "downstream transport of BC" here. I doubt that readers that are unfamiliar with MRIO analysis know what precisely what you mean here.

Reviewer #4 (Remarks to the Author):

I was satisfied with the author's present response.

We thank the reviewers for their constructive comments on our manuscript. In the revised version, major efforts have been elaborated to address all the reviewers' concerns and highlight the novelty. We hope the revised manuscript is improved to meet the standards set up by the reviewers. Detailed responses are listed point by point below in black font, with the comments in blue font.

Response to Anonymous Reviewer 2:

The article has undergone substantial improvement since the last revision, and I would like to thank the authors for their work and the thoughtful consideration of all of the reviewers' comments.

My comments were all addressed, and in my opinion the article reads much better. I do uphold my concerns about clear denominations of the different quantities that I already mentioned in my previous review. I have made a few specific detailed comments where these inconsistencies are apparent. I believe that the article still lacks a proficient use of the right denominations for each quantity that is discussed.

Response:

We really appreciate the reviewer's positive comments on our efforts in improving the manuscript. We have provided more clarifications to the contribution of this paper and made the denominations clearer. For example, we explain the concepts of different quantities (i.e., the production-based emission and consumption-based emission) in the Introduction section and rephrase the relevant sentences by using more general words to facilitate understanding for readers that are unfamiliar with MRIO analysis. We also clarify the results from production or consumption perspective and we believe that these modifications substantially help to avoid the confusion among different quantities.

Please refer to our response to the specific comments below and the revised manuscript for more details.

Detailed responses are listed point by point below.

While I think that at this point the article is on track to be ready for publication, I do suggest that the authors publish the article in a different journal. The basic take-away message from the Discussion section is that those countries that are located close to the HTP (and that would suffer the most from the glacial retreat) bear the main responsibility for the black carbon emissions above the HTP.

This is a result that has been known before (and the authors also discuss this in the beginning). The recommendations are focused on improving combustion efficiency in the neighbouring countries of the HTP, which is an outcome that does not require extensive trade-link analyses. The discussion also features that the trade-related impacts on BC over the HTP are higher than the direct pathways of emissions originating within the trade partners' border. But this is more of a qualitative statement, and it is also not particularly surprising given the short lifespan to BC in the atmosphere. A quantitative assessment of direct impacts/trade related impacts and the implication of these in the recommendations is not included in the Discussion section.

Therefore, I cannot see the strong research contribution or outcome in this paper that would in my view justify a publication in Nature Communications.

Response:

In this study, we quantify the contributions from different countries/sectors to the BC pollution over the HTP glaciers along the global production chain, from both production and consumption perspectives. The BC pollution and climate effect over the HTP glaciers have typically been regarded as a regional problem within the Asian continent. The major contribution of this study is that we reveal **the role of global trade in aggravating this problem by a virtual transport of BC from distant regions to the HTP glaciers and shed light on the different contribution of the regions as producers and consumers**. We also improved the Discussion section by adding the quantitatively results and clarify the valuable implications of this study (see L232-257, P6-7): 1) Asian countries/regions located close to the HTP should devote substantial efforts to minimize their BC emissions as producers, especially in India and China. 2) distant countries/regions, which was not identified previously, should be cautious in selecting their trading partners and share the responsibilities in the regulations of trade-related emissions (subsidize cleaner production funds/technologies to the HTP surrounding countries) by considering the potential impacts on the HTP glaciers.

The detailed clarification of novelty and major contributions, as well as the corresponding discussions in the text are as shown as below:

- (1) **We for the first time quantify the contributions from different countries/sectors, as both the producers and consumers, to the BC pollution over the HTP glaciers along the global production chain.** Previous studies focused on the source attribution of the BC pollution over the HTP have only considered the BC emissions released by the producers but ignored the potential effects of the consumers, who drive and outsource, with an increasing trend, the emissions to other regions via global trade (Kopacz et al., 2011 ; Lu et al., 2012 ; Zhang et al., 2015). The efficiency of virtual transport of BC via global trade

is orders of magnitude higher than atmospheric transport. For instance, regions like USA and Europe exert nearly 10 times larger effects on the HTP as consumers in global trade than their direct contributions through atmospheric transport pathways. The interregional BC flow embodied in global trade can directly elevate the BC burden over the HTP and cause increasing warming effects, which accelerates the melting of HTP glaciers. Our results show nearly 10 % of BC over the HTP glaciers is related to the effects of global trade through the relocation of production activities and exchange of commodities. This fraction could increase dramatically in the near future if more labor-intensive and energy-intensive production activities are outsourced to the HTP surrounding countries (especially India) without any international regulation.

(2) **We shed light on the future potential efforts in controlling cross-boundary BC pollution over the HTP glaciers.** Our analysis reveals that the consumption-based contribution from India to BC over the HTP glaciers has substantially increased by about 40 % from 2000 to 2014. Meanwhile, BC emissions in India that are related to goods and services consumed in other regions have been doubled over the last decade. This emission is projected to increase in the future because of change in global trade structure. The rise of trade among developing countries may continually increase the contribution from India to BC over the HTP glaciers. Present international policies created to deal with the transboundary air pollutions, such as the Long-Range Transboundary Air Pollution (LRTAP) Convention and the Malé Declaration on Control and Prevention of Air Pollution, only consider the responsibilities of producers in emissions and the atmospheric transport of air pollutants, but ignore the virtual transport embodied in trade, which may super-efficiently transport of pollution to some environmental/climate vulnerable areas like HTP. The BC pollution over the HTP glaciers have typically been regarded as a regional problem within the Asian continent. Our study reveals that the BC pollution over the HTP glaciers is a global problem linking to international trade. Therefore, global efforts are needed for an effective mitigation of the BC pollution over the HTP glaciers. **We hope our paper will arouse international environmental policymakers to aware this virtual long-range transport problem.** The potential effect of international trade and the shared responsibilities of consumers should be considered in future international policies for the cross-border air pollution regulation. With these efforts, we believe not only pollution, but also advanced cleaner production technology and efficient air pollution management strategies will also be exported to the HTP surrounding countries. The international cooperation will have comprehensive benefits to HTP glaciers, human health and carbon mitigation over the world most populated region.

(3) **We integrate multi-models to trace the climate effect over HTP to the final consumers along the atmospheric transport and global supply chain.** The adjoint inverse model,

different from the traditional chemistry transport model (CTM), provides a source-receptor sensitivity quantification for BC over the HTP in a model grid level. It traces the BC pollution over the HTP backward to regions where the direct geographic emission occurs. The multi-region input-output (MRIO) analysis further traces the territorial emissions in the production of products or services to regions where the related goods and services are ultimately consumed. **This study is a first time to combine the MRIO model with the adjoint inverse model, which can therefore completely assess the contributions of a specific country/region to BC pollutions over the HTP from various perspectives and identify the essential driving factors of these pollutions.** This approach will be a useful tool for future international policymaking regarding the full perspective of transboundary air pollution problem.

The vulnerability of HTP has been a focus of climate change, because of its significance to the water availability and food security for hundreds of millions of people (Immerzeel et al., 2010 ; Menon et al., 2010 ; Kraaijenbrink et al., 2017 ; Pritchard, 2017). An efficient mitigation of BC is urgently needed to preserve the HTP glaciers. **The multi-disciplinary perspectives can provide complementary policy insights into future mitigation policies.** We believe our paper is extremely salient and will be of interest to a wide range of scientists, policymakers, and the general public. We believe our manuscript well matches the scope and satisfies the criteria of *Nature Communications*.

Page 6-7 Line 232-241

“Although emissions produced in Asian countries/regions, especially India and China, play crucial roles on the BC pollution over the HTP glaciers due to their high source-receptor sensitivities to the HTP glaciers, our study reveals global trade can further aggravate this problem through a virtual transport pathway of BC from distant regions like US and Europe to the HTP glaciers. Nearly 10 % of BC over the HTP glaciers can be attributed to global trade through the relocation of production activities and exchange of commodities. The rapid rise of trade among developing countries contributes to the increase of BC emitted in India and transported to the HTP, which increasingly threatening the HTP glaciers.”

Page 7 Line 243-252

“Given the significant effects of international trade on BC pollutions over the HTP glaciers, collaborative efforts are needed for an effective mitigation of the BC pollution over the HTP glaciers. Asian countries, especially India and China, should expend substantial efforts, such as improving their energy efficiency, developing clean coal technologies, and promoting clean energy sources, to minimize their domestic BC emissions. Countries/regions that primarily import commodities should care more about the selection of their trading partners or provide assistance in the regulations of trade-related emissions to mitigate corresponding

climate effects because the sensitivities of BC over the HTP glaciers to various source regions show large differences. ”

Page 7 Line 254-263

“Present international policies to regulate the transboundary air pollutions, such as the Long-Range Transboundary Air Pollution (LRTAP) Convention³⁴ and the Malé Declaration on Control and Prevention of Air Pollution³⁵, consider only the responsibilities of direct emitters and physical transboundary air pollutants. The potential effect of international trade on air pollutions and the shared responsibilities of consumers in this issue are generally ignored by policy-makers. The virtual transport of air pollutants related to the trade of goods and services among countries/regions can be orders of magnitude larger than the typical atmospheric transport²⁵. Therefore, future policies should consider the effects of international trade to preserve vulnerable regions like the HTP glaciers.”

Detailed review:

Line 124:

I assume that “relative contribution from China” relates to the production-based emissions. I would suggest you use consistent naming of the quantities.

Thanks for your suggestion! We have rephrased this sentence (Line 126, P4):

“The contribution of BC emissions produced in China (including the HTP) to the BC pollution over the HTP increased from 17.4 % in DJF to 33.4 % in JJA.”

Line 147:

You are using the phrase “downstream transport of BC” here. I doubt that readers that are unfamiliar with MRIO analysis know what precisely what you mean here.

We have revised this sentence:

“The consumption of Middle Asia enables the transfer of BC emissions (0.06 Tg) to primarily China and Europe through the import of goods and services. ”

References:

- Immerzeel Walter W, Van Beek Ludovicus PH, Bierkens Marc FP. 2010. Climate change will affect the Asian water towers[J]. *Science*, 328(5984): 1382-1385.
- Kopacz M., Mauzerall D. L., Wang J., et al. 2011. Origin and radiative forcing of black carbon transported to the Himalayas and Tibetan Plateau[J]. *Atmospheric Chemistry and Physics*,

11(6): 2837-2852.

Kraaijenbrink P. D. A., Bierkens M. F. P., Lutz A. F., et al. 2017. Impact of a global temperature rise of 1.5 degrees Celsius on Asia's glaciers[J]. *Nature*, 549(7671): 257-260.

Lu Zifeng, Streets David G., Zhang Qiang, et al. 2012. A novel back-trajectory analysis of the origin of black carbon transported to the Himalayas and Tibetan Plateau during 1996-2010[J]. *Geophysical Research Letters*, 39(1): n/a-n/a.

Menon Surabi, Koch Dorothy, Beig Gufran, et al. 2010. Black carbon aerosols and the third polar ice cap[J]. *Atmospheric Chemistry and Physics*, 10(10): 4559-4571.

Regional Resource Center for Asia and the Pacific. Malé Declaration on Control and Prevention of Air Pollution. <http://www.rrcap.ait.asia/male>.

Pritchard Hamish D. 2017. Asia's glaciers are a regionally important buffer against drought[J]. *Nature*, 545(7653): 169-174.

The United Nations Economic Commission for Europe (UNECE). Convention on Long-range Transboundary Air Pollution. http://www.unece.org/env/lrtap/lrtap_h1.html.

Zhang R., Wang H., Qian Y., et al. 2015. Quantifying sources, transport, deposition, and radiative forcing of black carbon over the Himalayas and Tibetan Plateau[J]. *Atmospheric Chemistry and Physics*, 15(11): 6205-6223.

Response to Anonymous Reviewer 4:

I was satisfied with the author's present response.

Response:

We thank the authors for their constructive comments and the satisfaction with the improvements.

REVIEWERS' COMMENTS:

Reviewer #2 (Remarks to the Author):

Suggested outcome: no further revision needed

I would like to thank the authors for their efforts in revising the manuscript. The responses to my questions were insightful and have outlined in detail what the novelty of this research is.